# Pathological and immunological evaluation of different regimens of praziquantel treatment in a mouse model of *Schistosoma mansoni* infection

Ulrich Membe Femoe[1,2,3], Hermine Boukeng Jatsa[1,3]*, Valentin Greigert[2], Julie Brunet[2], Catherine Cannet[4], Mérimé Christian Kenfack[1,3], Nestor Gipwe Feussom[1,3], Joseph Bertin Kadji Fassi[1,3], Emilenne Tienga Nkondo[1,3], Ahmed Abou-Bacar[2], Alexander Wilhelm Pfaff[2], Théophile Dimo[1], Pierre Kamtchouing[1], Louis-Albert Tchuem Tchuenté[5,3]

1 Laboratory of Animal Physiology, Department of Animal Biology and Physiology, Faculty of Science, University of Yaoundé I, Yaoundé, Cameroon, 2 Institute of Parasitology and Tropical Diseases, UR 7292, Dynamic Host-Pathogen Interactions, University of Strasbourg, Strasbourg, France, 3 Centre for Schistosomiasis and Parasitology, Yaoundé, Cameroon, 4 Laboratory of Histomorphometry, Institute of Legal Medicine, University of Strasbourg, Strasbourg, France, 5 Laboratory of Parasitology and Ecology, Department of Animal Biology and Physiology, Faculty of Science, University of Yaoundé I, Yaoundé, Cameroon

* mjatsa@yahoo.fr

**Data Availability Statement:** All relevant data are within the manuscript.

## Abstract

### Background

One of the considerable challenges of schistosomiasis chemotherapy is the inefficacy of praziquantel (PZQ) at the initial phase of the infection. Immature schistosomes are not susceptible to PZQ at the curative dose. Here, we investigated the efficacy of different PZQ regimens administered during the initial stage of *Schistosoma mansoni* infection in mice.

### Methodology/Principal findings

Two months-old mice were individually infected with 80 *S. mansoni* cercariae and divided into one infected-untreated control group (IC) and four PZQ-treated groups: PZQ at 100 mg/kg/day for five consecutive days (group PZQ1), PZQ at 100 mg/kg/day for 28 days (group PZQ2), PZQ at 18 mg/kg/day for 28 days (group PZQ3) and a single dose of PZQ at 500 mg/kg (group PZQ4). The treatment started on day one post-infection (*p.i*), and each group of mice was divided into two subgroups euthanized on day 36 or 56 *p.i*, respectively. We determined the mortality rate, the parasitological burden, the hepatic and intestinal granulomas, the serum levels of Th-1, Th-2, and Th-17 cytokines, and gene expression. The treatment led to a significant ($p < 0.001$) reduction of worm burden and egg counts in the intestine and liver in groups PZQ2 and PZQ3. On 56[th] day *p.i*, there was a significant reduction ($p < 0.001$) of the number and volume of the hepatic granulomas in groups PZQ2 and PZQ3 compared to group PZQ1 or PZQ4. Moreover, in group PZQ3, the serum levels of IFN-γ, TNF-α, IL-13, and IL-17 and their liver mRNA expressions were significantly reduced

**Funding:** A part of this work has been conducted under the financial supports of the France Government by the Cooperative and Cultural Action Service (SCAC) scholarship 959139G and the Strasbourg Institute of Parasitology and Tropical Diseases (IPPTS) to Ulrich Membe Femoe. The funders had no role in study design, data collection and analysis, decision to publish, or preparation of the manuscript.

**Competing interests:** The authors have declared that no competing interests exist.

while IL-10 and TGF-β gene expression significantly increased. The highest mortality rate (81.25%) was recorded in group PZQ2.

## Conclusion/Significance

This study revealed that the administration of PZQ at 18 mg/kg/day for 28 consecutive days was the optimal effective posology for treating *S. mansoni* infection at the initial stage in a murine model.

## Author summary

According to WHO, more than 290.8 million people required preventive treatment against schistosomiasis in 2018. The pathology caused by the tissue-embolized eggs laid by adult worms is associated with symptoms such as anemia, tissue injuries, and severe fibrosis. Despite efforts to control this disease, it remains a public health problem in many tropical countries. One of the main challenges for managing this disease is to set up an efficient treatment at the early stage of the infection. Unfortunately, there is no appropriate therapeutic regimen for this purpose to the best of our knowledge. In this study, we propose an efficient praziquantel-based regimen that prevents the installation of the pathology in a mouse model of schistosomiasis. We found that praziquantel administration at the early stage of the infection prevents *Schistosoma mansoni*-infected mice from liver and intestine injuries, anemia, chronic inflammation, and severe fibrosis by reducing parasitic burden. Firstly, this study offers a reference treatment for further experimental investigations. Secondly, it could be a starting point for developing new protocols for preventive chemotherapy in endemic areas.

## Introduction

Schistosomiasis is an acute and chronic parasitic disease caused by *Schistosoma* blood flukes. *Schistosoma mansoni*, *S. japonicum*, *S. haematobium*, *S. intercalatum*, *S. guineensis*, and *S. mekongi* are the primary species that infect humans. Estimates show that at least 290.8 million people required preventive treatment in 2018, and about 700 million persons are at risk of infection in 78 countries where schistosomiasis is endemic [1]. In the course of schistosomiasis infection, the immune response progresses through at least three phases. In the first 3–5 weeks, during which the host is exposed to migrating immature parasites, the dominant response is T helper 1 (Th1)-like. As the parasites mature, mate and begin to produce eggs at weeks 5–6, the response alters markedly; the Th1 component decreases and is associated with the emergence of a strong Th2 response. This response is induced primarily by egg antigens and is characterized by tissues fibrosis around granulomatous response. Finally, during the chronic phase of the infection, the Th2 response is modulated, and granulomas that form around newly deposited eggs are smaller than at earlier times during infection [2–4].

After more than 40 years of use, praziquantel (PZQ) remains the drug of choice for the treatment of schistosomiasis. Its good levels of tolerance and safety, excellent acceptability by patients, easy administration, good efficacy profile against adult worms of all human *Schistosoma* species, and low cost led to its widespread use [5–7]. Despite the global efforts to control this disease, its morbidity and mortality are still significant in tropical and subtropical regions,

with more than 300 000 deaths per year. One of the main challenges of schistosomiasis chemotherapy is its efficacy at the initial phase of the infection [8].

Several authors have already demonstrated that immature *S. mansoni* worms are less susceptible to chemotherapeutic action at the initial stage of development, including oxamniquine and PZQ [9–14]. Sabah et al. [10] demonstrated that *S. mansoni* immature worm is less susceptible to at least six antischistosomal drugs used at curative doses against adult worms: hycanthone, oxamniquine, niridazol, amoscanate, antimonium-potassium artesunate, and PZQ. Treatment of infection at the initial stages would bring an excellent advantage for vertebrate hosts since it would avoid the production of eggs and, consequently, the formation of granulomas, which are the fundamental elements of the pathology of the disease [8,15,16]. Since the mouse model of schistosomiasis mansoni is the most used model for evaluating antischistosomal potencies of molecules, this study was conducted to determine the optimal posology of PZQ for the early stages treatment of schistosomiasis in a murine model. In addition to the parasitological burden, we evaluated the Th1/Th2 and Th17 mediated response in *S. mansoni*-infected mice treated with PZQ. For studies on the murine model of schistosomiasis, two curative regimens are recommended: a single oral dose of 500 mg/kg [17] or a dose of 100 mg/kg orally administered for 5 consecutive days [18].

## Methods

### Ethics statement

In this study, all procedures performed on mice followed the internationally accepted principles for laboratory animal use and care of the European Community guidelines (EEC Directive of 1986; 86/609/EEC). The protocol was approved by the Animal Ethics Committee of the Laboratory of Animal Physiology of the Faculty of Sciences, University of Yaoundé I–Cameroon.

### Animals

This study used two months old male BALB/c mice from the animal house of the Centre for Schistosomiasis and Parasitology (Yaoundé, Cameroon), weighing 22–27g. They were housed in polypropylene cages with free access to rodent diet and water and maintained under controlled conditions (temperature between 22 and 25˚C; 12-hour light and dark cycle). After that, 80 mice were individually infected with 80 *S. mansoni* cercariae using the method of tail and legs immersion as previously described [19]. Cercariae were harvested from naturally infected snail intermediate hosts *Biomphalaria pfeifferi*, collected at the river "Mock" (Makenene, Cameroon), by exposure to light for 3 hours.

After infection, mice were randomly divided into two batches. Each batch comprising forty mice was divided into five groups with eight mice each:

- IC: Infected-untreated mice receiving distilled water at the dose of 10 mL/kg;

- PZQ1: Infected mice treated with PZQ at 100 mg/kg/day from the 1st day post-infection (*p.i*) for 5 consecutive days [18];

- PZQ2: Infected mice treated with PZQ at 100 mg/kg/day from the 1st-day *p.i* for 28 days;

- PZQ3: Infected mice treated with PZQ at 18 mg/kg/day from the 1st-day *p.i* for 28 days. This daily dose was obtained by dividing 500 mg/kg by 28 days;

- PZQ4: Infected mice treated with a single oral dose of PZQ at 500 mg/kg on the 1st-day *p.i* [5,17].

A group of 6 uninfected-untreated named HC mice were added to each batch and used as a control.

PZQ was administered by gavage, as Cesol tablets (Merck KGaA, Darmstadt, Germany), ground into powder, and dissolved in distilled water. The mice of the first batch were sacrificed by cervical dislocation on the 36th-day *p.i* to evaluate the efficacy of PZQ during the acute phase of infection. Mice of the second batch were sacrificed on the 56th-day *p.i* to appreciate the protective effect of PZQ on the granulomatous stage of infection.

## Measurement of the body and organs indices

Each mouse was weighed weekly during the experiment to assess the body weight variation. After sacrifice, we removed the liver, spleen, thymus, and intestine from each mouse, weighed them and calculated their relative weights (g of organ /100 g of body weight).

## Harvesting of adult worms from the portal system

To assess the parasite burden, we recovered worms from each infected mouse using the perfusion technique described by Duvall and Dewitt [20]. After that, we counted them under a stereo-microscope. Finally, we calculated the percentage of reduction of worm number as follows:

$$P = \left( \frac{C - V}{C} \right) * 100$$

P = percentage of reduction; C = mean number of worms recovered from infected-untreated mice; V = mean number of worms recovered from infected-treated mice.

## Egg count in the feces, the liver, and the intestine

Feces from each infected mouse were collected before each sacrifice and weighed. After homogenization in 10% buffered formaldehyde, the mixture was stored at room temperature, and two aliquots of 100 µL each were counted on a light microscope. After sacrifice, the liver's left lobe and intestine were removed, weighed, and digested separately in 4% KOH as described in our previous study [19]. Using a light microscope, the number of eggs in each sample was determined in two aliquots of 100 µL each. Results were expressed as the mean number of eggs per gram of feces or tissue for the liver and intestine. The egg retention in tissues was the number of eggs trapped in the liver and the intestine and expressed as eggs per gram.

## Hematological analysis

Before sacrifice, we collected the blood of each mouse from retro-orbital plexus into EDTA tubes for the hematological analysis conducted with a Sysmex XP-3000 Hematology Analyzer (Sysmex Corporation, Kobe, Japan).

## Evaluation of some liver function biomarkers

A part of the blood was also collected in dry tubes and centrifuged at 3500 rpm for 15 min at 4˚C. The serum obtained was stored at -70˚C and used for the measurement of some parameters of the liver function. First, we determined the total proteins level using the method of Biuret [21], and the amount of protein was calculated from a standard curve using serial concentrations of bovine serum albumin. We also assessed alanine aminotransferase (ALT) and aspartate aminotransferase (AST) activities using Biolabo kits (Maizy, France) and according to the method described by Reitman and Frankel [22]. In addition, we measured alkaline phosphatase (ALP) and gamma-glutamyl transferase (γ-GT) activities using Biolabo kits (Maizy,

France) and according to the method of Wu [23]. Finally, we assayed total bilirubin according to the protocol described in the Inmesco kit (Wied, Germany) and read the red color developed after bilirubin's reaction with sulfanilic acid at 546 nm against a blank.

## Evaluation of some oxidative stress biomarkers

The effect of PZQ treatment on the hepatic and splenic oxidative stress was evaluated as follows: after sacrifice, 0.4 g of the right lobe of the liver was homogenized in 2 mL of a Tris-HCl 50 mM buffer, pH 7.4, while 0.2 g of the spleen was homogenized in 400 μL of the same buffer. The homogenates (20% for liver and 50% for spleen w/v) were centrifuged at 3000 rpm for 25 min at 4˚C, and the supernatant was collected and stored at -70˚C until assayed.

We first measured the lipid peroxides as thiobarbituric acid reactive substances. The product of the reaction between malondialdehyde (MDA) and thiobarbituric acid was estimated as described by Wilbur et al. [24]. The concentration of MDA was calculated using a molar extinction coefficient of $1.56 \times 10^{-5}$ mmol$^{-1}$cm$^{-1}$, and results were expressed as nmol/g of the liver. In addition, we determined nitrite concentration using the Griess reagent (1% N-1-naphtylethylene diamine + 1% sulfanilamide in 2.5% phosphoric acid) as described by Hibbs et al. [25]. Nitrite levels were calculated from a standard curve using serial concentrations of sodium nitrite (1 mM) and expressed as μmol/g of the liver. We also estimated superoxide dismutase (SOD) activity according to the protocol described by Misra et Fridovish [26]. SOD activity was expressed as SOD units/g of the liver, and one unit of SOD is defined as the enzyme activity that inhibits 50% adrenaline oxidation to adrenochrome. Moreover, we used the Ellman reagent (5 mg of 2, 2-dithio-5, 5-nitrobenzoic acid + 250 mL phosphate buffer) to measure the concentration of reduced glutathione [27]. We finally determined catalase activity from a standard curve as described by Sinha [28] and expressed it as mol/min/g of the liver.

## Histology and morphometric analysis of the liver and intestinal inflammatory granulomas

A portion of each animal's right liver lobe and ileum was immersed in 10% formaldehyde prepared in phosphate-buffered saline (PBS). After dehydration in successive ethanol baths (50˚C to 100˚C) and xylene, tissue samples were paraffin-embedded. Five micrometer thick sections were stained with hematoxylin and eosin (H&E) for qualitative analysis of the liver and the granulomatous inflammatory reaction evaluation. The second set of slides was stained with picrosirius (PS) to highlight liver and ileal fibrosis characterized by collagen deposition around granuloma. Images were acquired at x10 and x20 using CMOS Digital Camera (IDS, Germany) connected to an optical microscope (Axiophot, Zeiss, Germany) and analyzed by PathScan Touch software (Excilone, France). For quantitative measurement of liver and intestine granulomas, images were captured with an x10 objective at an x10 ocular magnification with a digital camera AmScope MD 500 (AmScope, USA) connected to a wild-type optical microscope and analyzed with ImageJ 1.32 software (NIH, USA). All granulomas containing a single central egg with miracidia were selected for each mouse. Granulomas in which it was possible to observe more than one egg or granulomas in which the egg was not visible or destroyed were excluded from the analysis. The number of granulomas, as well as their volume, were determined. All granulomas per mouse were measured using the largest diameter and perpendicular diameter. The volume of each granuloma was calculated assuming a spherical shape using the following formula:

$$V = \frac{4(\pi * R^3)}{3}$$

The radius R was obtained by dividing the main diameter of the lesion by two. The mean granuloma volume for each group was calculated [29,30].

## Serum levels of cytokines

According to the manufacturer's recommendations, we used the mouse ProtocartaPlex 10-plex immunoassay kit (Thermofisher Scientific, Vienna, Austria) to measure cytokine levels in serum samples. We classified the immune mediators into five categories: proinflammatory mediator (Monocyte Chemoattractant Protein [MCP]–1), type 1 cytokines (IL-2, interferon [IFN]- γ, and tumor necrosis factor [TNF]–α), type 2 cytokines (IL-4, IL-5, IL-10, and IL-13), a T helper 17 cytokine (IL-17), and a regulatory cytokine (Transforming Growth Factor [TGF]-β1). Data were acquired on the MAGPIX-Luminex XMAP technology (Merck Milli-pore, Darmstadt, Germany) using the Luminex XPONENT software solutions. The concentra-tion (pg/mL) of each cytokine was obtained by interpolating MFI (median fluorescent intensity) of a dilution standard curve over seven dilution points supplied with the kit and cal-culated by the ProtocartaPlex Analyst 1.0 software (eBioscience, Thermo Fisher Scientific, USA).

## Real-time quantitative PCR analysis

According to the manufacturer's instructions, we extracted the total RNA from 10 mg of liver tissue using NucleoZOL reagent (Macherey-Nagel GmbH & Co. KG, Germany). After that, we determined the RNA concentration using the NanoDrop 2000c spectrophotometer (Thermo Scientific, USA). Then, each RNA sample was reversely transcribed into cDNA (qScript cDNA Synthesis kit, Quanta Biosciences, Inc, USA). Next, we mixed the obtained cDNA with SsoAd-vanced Universal SYBR Supermix kit (Bio-Rad, USA). We then amplified it using CFX Con-nect Real-Time System (Bio-Rad, USA), with glyceraldehyde 3-phosphate dehydrogenase (GAPDH) as a reference gene. Finally, we calculated the expression of each cDNA using a standard curve, and the relative expression of mRNA in each sample was expressed as the GAPDH ratio. Primers purchased from Eurofins Genomics (Nantes, France) and used for RT-qPCR are shown in Table 1. All these primers were synthetized according to the PrimerBank online database (https://pga.mgh.harvard.edu/primerbank/citation.html).

## Statistical analysis

Results are expressed as means ± SEM. Statistical differences between groups were evaluated with a Student t-test or one-way ANOVA followed by the Tukey multiple comparison post-

**Table 1. Mice primers used for RT-qPCR.**

| mRNA | Forward Primer (5'→ 3') | Reverse primer (5'→ 3') | Amplicon size |
|---|---|---|---|
| GAPDH | AGGTCGGTGTGAACGGATTTG | TGTAGACCATGTAGTTGAGGTCA | 123 |
| CXCL-10 (IP10) | CCAAGTGCTGCCGTCATTTTC | TCCCTATGGCCCTCATTCTCA | 133 |
| MCP-1 (CCL2) | TTAAAAACCTGGATCGGAACCAA | GCATTAGCTTCAGATTTACGGGT | 121 |
| MIP-1α (CCL3) | TTCTCTGTACCATGACACTCTGC | CGTGGAATCTTCCGGCTGTAG | 100 |
| RANTES (CCL5) | GCTGCTTTGCCTACCTCTCC | TCGAGTGACAAACACGACTGC | 104 |
| IFN-γ | ACAGCAAGGCGAAAAAGGATG | TGGTGGACCACTCGGATGA | 106 |
| IL-10 | GCTCTTACTGACTGGCATGAG | CGCAGCTCTAGGAGCATGTG | 105 |
| IL-13 | CCTGGCTCTTGCTTGCCTT | GGTCTTGTGTGATGTTGCTCA | 116 |
| FGF1 | CCCTGACCGAGAGGTTCAAC | GTCCCTTGTCCCATCCACG | 122 |
| TGF-β1 | CTCCCGTGGCTTCTAGTGC | GCCTTAGTTTGGACAGGATCTG | 133 |

test. Analyses were performed using GraphPad Prism version 8.0.1 (GraphPad Software, San Diego, CA, USA). A *p*-value of <0.05 was considered statistically significant.

## Results

### Mice survival rate after early administration praziquantel regimens to *Schistosoma mansoni*-infected mice

During the experimentation, we recorded no death in uninfected mice (HC), infected-untreated mice (IC), infected-treated mice with 18 mg/kg of PZQ for 28 days (PZQ3), and a single dose of 500 mg/ kg of PZQ (PZQ4) (Fig 1). We recorded the highest death rate in the group of mice treated with 100 mg/kg of PZQ for 28 days (PZQ2), where survival rates were 85.71%, 42.86%, and 18.75% at 6, 7, and 8 weeks respectively. The survival rate was 53.57% in the group of mice treated with PZQ 100 mg/kg for 5 days (PZQ1).

### Praziquantel regimens limit the body weight loss in *Schistosoma mansoni*-infected mice

At the end of the acute phase of infection, no significant difference was observed in the body weight variation of uninfected mice (HC), infected-untreated mice (IC), and PZQ-treated mice (Fig 2A). In contrast, during the granulomatous phase from day 48 to day 56 post-infection, infected-untreated mice and PZQ-treated mice of groups PZQ1, PZQ2, and PZQ4 gradually lost body weight. Only mice from group PZQ3 treated with 18 mg/kg of PZQ for 28 days gained weight. Body weight decreased significantly ($p < 0.001$) in groups IC, PZQ1, and PZQ4 compared to PZQ3. No statistical difference was recorded between the weight gain of PZQ3 mice and that of uninfected mice (HC) (Fig 2B).

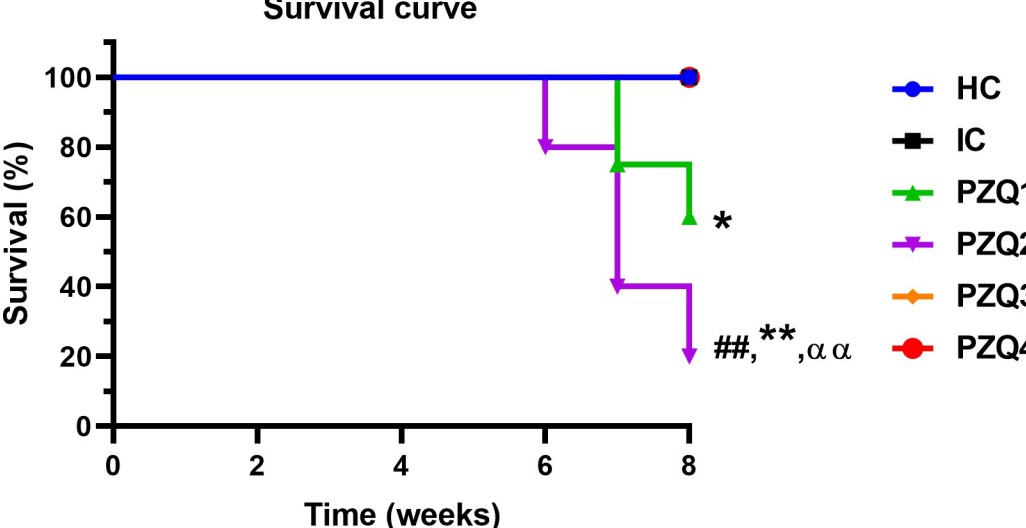

**Fig 1. Survival curve of mice infected with *Schistosoma mansoni* and treated with different praziquantel regimens.** HC: Uninfected-untreated mice, IC: Infected-untreated mice, PZQ1: Infected mice treated with praziquantel at 100 mg/kg for 5 consecutive days, PZQ2: Infected mice treated with praziquantel at 100 mg/kg for 28 consecutive days, PZQ3: Infected mice treated with praziquantel at 18 mg/kg for 28 consecutive days, PZQ4: Infected mice treated with a single dose of praziquantel at 500 mg/kg. ANOVA followed by Tukey multiple comparison test was also used for statistical analysis. ##p < 0.01: values are significantly different from those of uninfected mice (group HC). *p < 0.05, **p < 0.01: values are significantly different from those of infected-untreated mice (group IC). ααp < 0.01: values are significantly different from those of infected mice treated with PZQ3 posology (group PZQ3).

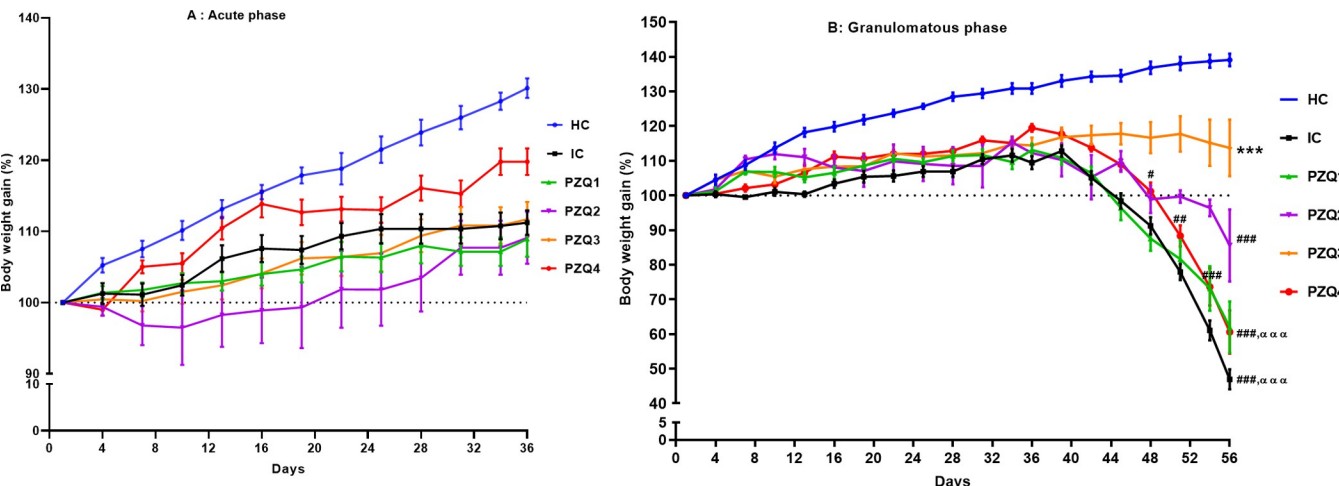

**Fig 2. Effect of praziquantel treatment on the body weight during acute (A) and granulomatous (B) stages of *Schistosoma mansoni* infection in mice.** HC: Uninfected mice, IC: Infected-untreated mice, PZQ1: Infected mice treated with praziquantel at 100 mg/kg for 5 consecutive days, PZQ2: Infected mice treated with praziquantel at 100 mg/kg for 28 consecutive days, PZQ3: Infected mice treated with praziquantel at 18 mg/kg for 28 consecutive days, PZQ4: Infected mice treated with a single dose of praziquantel at 500 mg/kg. Data are expressed as mean ± SEM (n = 6). ANOVA followed by Tukey multiple comparison test was also used for statistical analysis. [#]p < 0.05, [##]p < 0.01, [###]p < 0.001: values are significantly different from those of uninfected mice (group HC). [***]p < 0.001: values are significantly different from those of infected-untreated mice (group IC). [ααα]p < 0.001: values are significantly different from those of infected mice treated with PZQ3 posology (group PZQ3).

## Praziquantel regimens limit the enlargement of organs in *Schistosoma mansoni*-infected mice

During the acute phase of the infection, the spleen was the only organ showing a size increase with an index significantly increased by 47.52% in IC mice compared to HC mice. In contrast, no statistical difference was recorded between IC and HC mice or between the different PZQ treated groups. Notably, among PZQ-treated groups, when compared to IC mice, a significant decrease of the spleen index in PZQ2 ($p < 0.05$), PZQ3 ($p < 0.001$), and PZQ4 ($p < 0.05$) was observed. At the granulomatous stage of the infection (from day 36[th] to day 56[th] post-infection), the body weight loss of IC mice was associated with a significant increase of the liver, intestine, spleen, and thymus indices by 56.80%, 88.49%, 64.25%, and 68.57%, respectively in comparison to HC mice. In contrast, PZQ treatment at an 18 mg/kg dose for 28 days (PZQ3) almost wholly reversed this infection-induced increase. Moreover, there was no statistical difference in the liver and thymus indices of PZQ1, PZQ2, and PZQ4 mice compared to IC mice. Regarding the spleen index, its level was still increased in mice of PZQ1 (59.34%), PZQ2 (45.18%), and PZQ4 (38.84%) groups in comparison to that PZQ3 group. Results were similar regarding the thymus, which was significantly increased in PZQ1 ($p < 0.05$) and PZQ4 ($p < 0.01$) mice groups (Table 2).

## Praziquantel regimens reduce *Schistosoma mansoni* adult worms and egg production in mice

Worm burdens and egg loads in the liver, intestine, and feces are shown in Fig 3. The number of worms recovered from infected-untreated mice at the end of the acute phase of infection was 42.14 ± 2.87 and 45.62 ± 3.56 at the end of the granulomatous phase. In the acute phase of the disease, treatment with PZQ leads to a significant reduction ($p < 0.001$) of worm burden in groups PZQ2 (97.65%) and PZQ3 (97.29%) compared to group IC. This reduction was 71.50% in PZQ2 and 84.04% in PZQ3 groups during the granulomatous phase. No significant

**Table 2. Effect of praziquantel treatment on the relative organ weight (g/100g body weight) of *Schistosoma mansoni*-infected mice.**

| Groups | Liver | Intestine | Spleen | Thymus |
|---|---|---|---|---|
| **Acute phase** | | | | |
| HC | 5.55 ± 0.25 | 5.25 ± 0.29 | 0.53 ± 0.04 | 0.23 ± 0.02 |
| IC | 5.02 ± 0.33 | 5.99 ± 0.27 | 1.01 ± 0.12[##] | 0.24 ± 0.02 |
| PZQ1 | 5.46 ± 0.39 | 4.83 ± 0.13 | 0.83 ± 0.08 | 0.23 ± 0.02 |
| PZQ2 | 5.65 ± 0.28 | 5.70 ± 0.31 | 0.68 ± 0.07* | 0.23 ± 0.02 |
| PZQ3 | 5.41 ± 0.20 | 6.32 ± 0.39 | 0.57 ± 0.04*** | 0.25 ± 0.02 |
| PZQ4 | 5.19 ± 0.21 | 5.00 ± 0.17 | 0.70 ± 0.05* | 0.34 ± 0.02[α] |
| **Granulomatous phase** | | | | |
| HC | 5.30 ± 0.33 | 4.56 ± 0.29 | 0.59 ± 0.05 | 0.11 ± 0.01 |
| IC | 12.27 ± 0.64[###] | 13.56 ± 0.61[###] | 1.65 ± 0.13[###] | 0.35 ± 0.03[##] |
| PZQ1 | 10.08 ± 0.48 | 16.01 ± 0.87[ααα] | 1.82 ± 0.08[αααα] | 0.36 ± 0.02 [α] |
| PZQ2 | 10.58 ± 1.42 | 10.10 ± 0.93*, [ααα] | 1.35 ± 0.18[α] | 0.30 ± 0.06 |
| PZQ3 | 7.77 ± 0.60** | 4.43 ± 0.40*** | 0.74 ± 0.06*** | 0.17 ± 0.01* |
| PZQ4 | 10.67 ± 0.93 | 10.84 ± 0.77*, [ααα] | 1.21 ± 0.12*, [α] | 0.36 ± 0.05[αα] |

Acute phase: mice euthanization at the 36[th] day post-infection

Granulomatous satge: mice euthanization at the 56[th] day post-infection

HC: Uninfected mice, IC: Infected-untreated mice, PZQ1: Infected mice treated with praziquantel at 100 mg/kg for 5 consecutive days, PZQ2: Infected mice treated with praziquantel at 100 mg/kg for 28 consecutive days, PZQ3: Infected mice treated with praziquantel at 18 mg/kg for 28 consecutive days, PZQ4: Infected mice treated with a single dose of praziquantel at 500 mg/kg. Data are expressed as mean ± SEM (n = 6). ANOVA followed by Tukey multiple comparison test was also used for statistical analysis. ##$p < 0.01$

###$p < 0.001$: values are significantly different from those uninfected mice (group HC).

*$p < 0.05$

**$p < 0.01$

***$p < 0.001$: values are significantly different from those of infected-untreated mice (group IC).

α$p < 0.05$

αα$p < 0.01$

ααα$p < 0.001$: values are significantly different from those of infected mice treated with PZQ3 posology (group PZQ3).

statistical difference was observed between PZQ1 and IC mice during both stages of infection, while a significant difference ($p < 0.01$) was found between PZQ4 and IC mice during the granulomatous phase (Fig 3IA and 3IIA). In addition, in both phases of infection, the egg burden was significantly lower ($p < 0.001$) in the liver and intestine of PZQ2 and PZQ3 mice compared to IC mice (Fig 3IB, 3IC, 3IIB and 3IIC).

Regarding egg retention in tissues, we recorded a significant reduction in PZQ2 ($p < 0.05$) and PZQ3 ($p < 0.01$) groups compared to the IC group during the acute phase of infection. During the granulomatous phase, this burden decreases in groups PZQ 1 ($p < 0.05$), PZQ2 ($p < 0.001$) and PZQ3 ($p < 0.001$) compared to IC group. However, this reduction was more important in PZQ3 than in PZQ1 ($p < 0.001$). The reduction of the egg retention in tissues was more relevant in PZQ3 than PZQ2 (99.82% vs. 42.29%) (Table 3).

Egg excretion in the feces after PZQ treatment was only significantly different between groups during the granulomatous phase of infection. Compared to infected-untreated mice (group IC), a reduction of 99.06% and 87.83% was recorded in PZQ2 mice ($p < 0.01$) and PZQ3 mice ($p < 0.05$) respectively. These results also show that the reduction of worm burden and egg load in the liver and intestine of mice belonging to the PZQ3 group was lowered ($p < 0.001$) when compared to those of PZQ1 and PZQ4 mice (Fig 3IID).

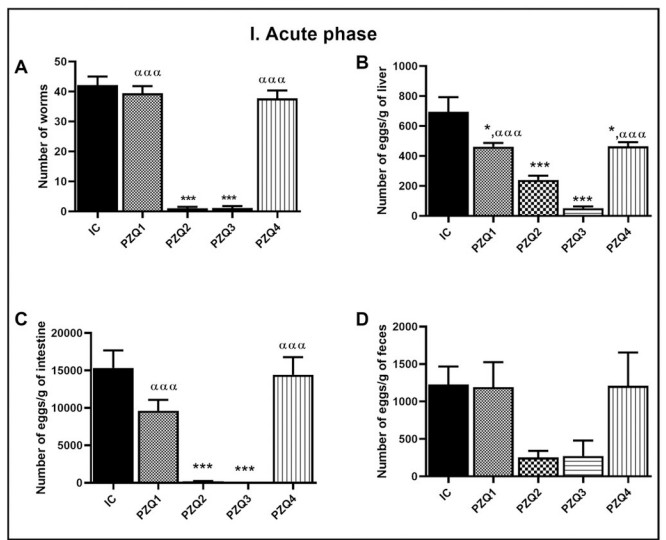
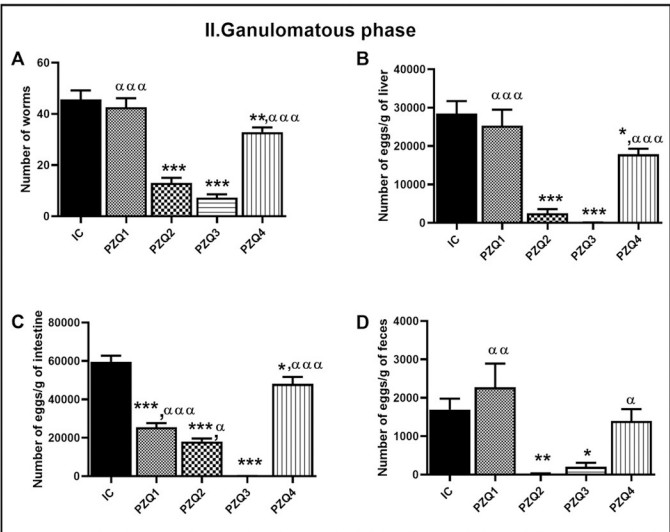

**Fig 3. Effect of praziquantel treatment on the worm burden (A) and the hepatic (B), intestinal (C), and fecal (D) egg loads on *Schistosoma mansoni* acute (I) and granulomatous (II) phase of infection in mice.** IC: Infected-untreated mice, PZQ1: Infected mice treated with praziquantel at 100 mg/kg for 5 consecutive days, PZQ2: Infected mice treated with praziquantel at 100 mg/kg for 28 consecutive days, PZQ3: Infected mice treated with praziquantel at 18 mg/kg for 28 consecutive days, PZQ4: Infected mice treated with a single dose of praziquantel at 500 mg/kg. Data are expressed as mean ± SEM (n = 6). ANOVA followed by Tukey multiple comparison test was also used for statistical analysis. *p < 0.05, **p < 0.01, ***p < 0.001: values are significantly different from those of infected-untreated mice (group IC). $^{\alpha}$p < 0.05, $^{\alpha\alpha}$p < 0.01, $^{\alpha\alpha\alpha}$p < 0.001: values are significantly different from those of infected mice treated with PZQ3 posology (group PZQ3).

## Praziquantel regimens protect against *Schistosoma*-induced anemia and leukocytes production in mice

*S. mansoni* acute infection induced a significant reduction of the level of red blood cells ($p < 0.001$), hemoglobin ($p < 0.05$), hematocrit ($p < 0.001$) and platelets ($p < 0.001$) in infected-untreated mice (IC) compared to uninfected ones (HC). Regarding the leucogram, a significant increase of the level of total white blood cells ($p < 0.01$) and granulocytes percentage ($p < 0.001$), but a decrease ($p < 0.001$) of lymphocytes percentage was recorded. These variations were more pronounced during the granulomatous stage of the infection. These changes were completely reversed in the PZQ3 group since there was no statistical difference between PZQ3 and HC groups. In contrast, treatment regimens of PZQ1 and PZQ4 groups

**Table 3. Effect of praziquantel treatment on egg retention in tissue in *Schistosoma mansoni*-infected in mice.**

| Infection stage | Groups | | | | |
|---|---|---|---|---|---|
| | IC | PZQ1 | PZQ2 | PZQ3 | PZQ4 |
| Acute | 8021.22 ± 2318.80 | 5044.36 ± 1448.29 | 217.13 ± 27.95* | 34.61 ± 7.65** | 7445.71± 2042.20$^{\alpha}$ |
| Granulomatous | 44002.13 ± 4573.33 | 25390.37 ± 2208.54*,$^{\alpha\alpha\alpha}$ | 10293.65 ± 3583.43*** | 76.84 ± 26.83*** | 32989.85 ± 3946.36$^{\alpha\alpha\alpha}$ |

IC: Infected-untreated mice, PZQ1: Infected mice treated with praziquantel at 100 mg/kg for 5 consecutive days, PZQ2: Infected mice treated with praziquantel at 100 mg/kg for 28 consecutive days, PZQ3: Infected mice treated with praziquantel at 18 mg/kg for 28 consecutive days, PZQ4: Infected mice treated with a single dose of praziquantel at 500 mg/kg. Data are expressed as mean ± SEM (n = 6). ANOVA followed by Tukey multiple comparison test was also used for statistical analysis.

*p < 0.05

**p < 0.01

***p < 0.001: values are significantly different from those of infected-untreated mice (group IC).

$^{\alpha}$p < 0.05

$^{\alpha\alpha\alpha}$p < 0.001: values are significantly different from those of infected mice treated with PZQ3 posology (group PZQ3).

**Table 4. Effect of praziquantel treatment on the hematological parameters on *Schistosoma mansoni*-infected mice.**

| Groups | RBC ($10^6$/mm$^3$) | Hemoglobin (g/dL) | Hematocrit (%) | Platelets ($10^3$/mm$^3$) | WBC ($10^3$/mm$^3$) | Lymphocyte (%) | Monocyte (%) | Granulocyte (%) |
|---|---|---|---|---|---|---|---|---|
| | | | | **Acute phase** | | | | |
| HC | 8.62 ± 0.39 | 14.03 ± 0.53 | 49.70 ± 1.67 | 525.67 ± 29.14 | 4.71 ± 0.36 | 72.53 ± 2.48 | 8.25 ± 0.24 | 19.21 ± 2.31 |
| IC | 4.56 ± 0.51[###] | 9.33 ± 0.76[#] | 28.38 ± 2.04[###] | 264.14 ± 25.10[###] | 10.33 ± 1.01[##] | 44.91 ± 4.72[###] | 10.60 ± 1.71 | 44.48 ± 4.64[###] |
| PZQ1 | 4.01 ± 0.51[αα] | 9.80 ± 1.42 | 31.60 ± 4.53 | 278.14 ± 31.61[ααα] | 10.01 ± 0.62 | 46.37 ± 2.99[ααα] | 10.44 ± 1.59 | 42.98 ± 1.62[ααα] |
| PZQ2 | 4.02 ± 0.50[###,αα] | 10.01 ± 1.16 | 32.77 ± 3.58 | 255.86 ± 18.65[###, ααα] | 11.78 ± 1.92[##,αα] | 44.93 ± 3.01[###,ααα] | 8.33 ± 0.94 | 46.34 ± 3.76[###,ααα] |
| PZQ3 | 7.19 ± 0.88[*] | 10.66 ± 0.65 | 37.61 ± 2.94 | 486.28 ± 11.19[***] | 5.94 ± 0.37[*] | 67.82 ± 2.29[***] | 11.65 ± 1.77 | 20.51 ± 0.62[***] |
| PZQ4 | 5.34 ± 0.41 | 10.57 ± 0.80 | 30.08 ± 2.49 | 214.89 ± 23.07[ααα] | 10.16 ± 0.68[αα] | 42.15 ± 2.26[ααα] | 6.20 ± 0.46 | 51.74 ± 1.77[ααα] |
| | | | | **Granulomatous phase** | | | | |
| HC | 9.77 ± 0.56 | 12.73 ± 0.93 | 55.51 ± 0.78 | 521.50 ± 16.78 | 4.45 ± 0.47 | 72.73 ± 0.96 | 10.26 ± 0.78 | 17.25 ± 1.52 |
| IC | 5.13 ± 0.33[###] | 10.03 ± 0.36 | 39.78 ± 2.12[###] | 233.50 ± 14.26[###] | 11.00 ± 0.58[###] | 32.42 ± 3.14[###] | 13.72 ± 1.30 | 53.85 ± 4.08[###] |
| PZQ1 | 6.17 ± 0.39[***,α] | 9.48 ± 0.42 | 47.86 ± 2.34 | 280.00 ± 22.91[ααα] | 10.54 ± 0.73[αα] | 39.74 ± 0.62[ααα] | 13.90 ± 1.02 | 46.36 ± 0.85[ααα] |
| PZQ2 | 9.99 ± 0.86[***] | 13.83 ± 1.33 | 53.00 ± 7.25[*] | 414.67 ± 22.67[*] | 11.50 ± 0.70[αα] | 68.76 ± 3.44[***] | 15.23 ± 1.03 | 16.00 ± 4.14[***] |
| PZQ3 | 8.20 ± 0.43[***] | 10.52 ± 0.67 | 50.18 ± 1.62[*] | 526.14 ± 50.04[***] | 5.32 ± 0.82[***] | 73.07 ± 5.03[***] | 16.18 ± 3.21 | 10.75 ± 2.00[***] |
| PZQ4 | 4.43 ± 0.30[ααα] | 10.58 ± 0.80 | 34.90 ± 2.48[ααα] | 272.10 ± 31.23[ααα] | 9.57 ± 0.77[ααα] | 28.65 ± 2.86[ααα] | 10.57 ± 1.04 | 60.78 ± 3.70[ααα] |

HC: Uninfected mice, IC: Infected-untreated mice, PZQ1: Infected mice treated with praziquantel at 100 mg/kg for 5 consecutive days, PZQ2: Infected mice treated with praziquantel at 100 mg/kg for 28 consecutive days, PZQ3: Infected mice treated with praziquantel at 18 mg/kg for 28 consecutive days, PZQ4: Infected mice treated with a single dose of praziquantel at 500 mg/kg. Data are expressed as mean ± SEM (n = 6). ANOVA followed by Tukey multiple comparison test was also used for statistical analysis.

#$p < 0.05$

##$p < 0.01$

###$p < 0.001$: values are significantly different from those uninfected mice (group HC).

*$p < 0.05$

***$p < 0.001$: values are significantly different from those of infected-untreated mice (group IC).

α$p < 0.05$

αα$p < 0.01$

ααα$p < 0.001$: values are significantly different from those of infected mice treated with PZQ3 posology (group PZQ3).

did not restore erythrogram, leucogram, and platelet count similar to those of the HC group. The efficacy of the PZQ2 scheme of treatment was observed in the granulomatous phase only (Table 4).

## Praziquantel regimens prevent liver injuries induced by *Schistosoma mansoni* infection in mice

To determine the impact of different protocols of PZQ treatment on *S. mansoni* infection-induced liver pathology, the activities of transaminases (ALT and AST), ALP, γ-GT, and total protein and total bilirubin concentrations, were measured in the serum. Results are shown in Table 5. During the acute stage of the infection (up to day 36[th] post-infection), the ALT ($p < 0.001$), AST ($p < 0.001$), and γ-GT ($p < 0.01$) activities in the infected control (IC) group were significantly increased, compared to those of the healthy control group (HC). PZQ treatment significantly decreased the activity of ALT in all treatment regimens ($p < 0.001$), while only the PZQ3 regimen led to a significant decrease ($p < 0.05$) of AST activity, as compared to IC mice. There was no statistical difference in the activity of ALP and the concentrations of total proteins and total bilirubin between any groups. At the granulomatous stage of the infection, infected-untreated mice (IC) had elevated activities of ALT (69.84%, $p < 0.001$), AST (75.90%, $p < 0.001$), ALP (46.45%, $p < 0.01$) and total bilirubin concentration (65.59%,

                    

**Table 5. Effect of praziquantel treatment on the liver function of *Schistosoma mansoni*-infected mice.**

| Groups | ALT (UI/L) | AST (UI/L) | ALP (UI/L) | γ-GT (UI/L) | Total bilirubin (μmol/L) | Total protein (μg/mL) |
|---|---|---|---|---|---|---|
| | | | **Acute phase** | | | |
| HC | 549.43 ± 60.45 | 329.21 ± 61.39 | 282.49 ± 69.73 | 144.93 ± 16.97 | 241.83 ± 16.97 | 2.07 ± 0.23 |
| IC | 1485.36 ± 117.83### | 1367.26 ± 96.26### | 246.80 ± 58.45 | 64.39 ± 4.97## | 349.55 ± 43.40 | 1.22 ± 0.14 |
| PZQ1 | 933.25 ± 39.42*** | 1057.23 ± 39.57 | 211.77 ± 27.00 | 93.93 ± 15.85 | 296.23 ± 39.30 | 0.98 ± 0.11 |
| PZQ2 | 1045.89 ± 98.44** | 1221.43 ± 109.09 | 214.88 ± 32.15 | 109.08 ± 15.90 | 366.85 ± 28.82 | 1.67 ± 0.24 |
| PZQ3 | 739.52 ± 20.13*** | 977.51 ± 97.98* | 161.16 ± 34.89 | 122.71 ± 12.27 | 277.09 ± 15.14 | 1.75 ± 0.23 |
| PZQ4 | 887.05 ± 76.07*** | 1119.23 ± 71.05 | 224.66 ± 59.66 | 101.92 ± 14.86 | 292.69 ± 19.42 | 1.35 ± 0.17 |
| | | | **Granulomatous phase** | | | |
| HC | 461.89 ± 58.47 | 421.95 ± 32.95 | 278.01 ± 55.32 | 159.96 ± 16.84 | 270.56 ± 41.09 | 2.23 ± 0.17 |
| IC | 1429.80 ± 51.42### | 1142.64 ± 76.61### | 519.11 ± 60.01## | 59.65 ± 11.05## | 786.19 ± 84.23### | 1.04 ± 0.13### |
| PZQ1 | 1430.08 ± 72.90αααα | 1415.96 ± 147.20 | 464.61 ± 59.34 | 97.56 ± 18.56 | 665.49 ± 68.80αα | 0.83 ± 0.89 |
| PZQ2 | 1730.68 ± 114.65αααα | 1735.48 ± 361.40*, αα | 512.30 ± 147.48 | 86.61 ± 1.76 | 388.11 ± 34.59** | 0.93 ± 0.13 |
| PZQ3 | 858.12 ± 127.86*** | 912.96 ± 72.59 | 280.28 ± 81.37* | 137.86 ± 25.84* | 295.67 ± 37.24*** | 1.31 ± 0.13 |
| PZQ4 | 1471.27 ± 51.50αααα | 1183.78 ± 99.23 | 485.05 ± 38.81α | 65.75 ± 8.18αα | 620.23 ± 37.98αα | 1.16 ± 0.21 |

HC: Uninfected mice, IC: Infected-untreated mice, PZQ1: Infected mice treated with praziquantel at 100 mg/kg for 5 consecutive days, PZQ2: Infected mice treated with praziquantel at 100 mg/kg for 28 consecutive days, PZQ3: Infected mice treated with praziquantel at 18 mg/kg for 28 consecutive days, PZQ4: Infected mice treated with a single dose of praziquantel at 500 mg/kg. Data are expressed as mean ± SEM (n = 6). ANOVA followed by Tukey multiple comparison test was also used for statistical analysis.

##p < 0.01

###p < 0.001: values are significantly different from those of uninfected mice (group HC).

*p < 0.05

**p < 0.001

***p < 0.001: values are significantly different from those of infected-untreated mice (group IC).

αp < 0.05

ααp < 0.01

αααp < 0.001: values are significantly different from those of infected mice treated with PZQ3 posology (group PZQ3).

$p < 0.001$) and decreased activities of γ-GT (62.71%, $p < 0.01$) and total protein concentration (53.36%, $p < 0.001$), compared to uninfected mice (HC). Treatment of infected mice with PZQ at the 18 mg/kg dose for 28 consecutive days (PZQ3) restored the transaminases (ALT and AST), ALP and γ-GT activities, and total protein, and total bilirubin concentrations in the serum at near-normal levels. In detail, ALT activity decreased by 39.98% ($p < 0.001$), ALP activity by 46.01% ($p < 0.05$) and total bilirubin level by 62.15% ($p < 0.001$), while γ-GT activity increased by 56.73% in the PZQ3 group compared to IC group. PZQ3 regimen was the most effective one since the other regimens did not show these beneficial effects on hepatic biomarker levels. Mice subjected to the PZQ2 regimen even showed the most elevated transaminases (ALT and AST) activities that were drastically higher than that of uninfected mice (HC) ($p < 0.001$).

## Praziquantel treatment preserves the hepatic and spleen oxidative balance in *Schistosoma mansoni*-infected mice

At the acute phase of the infection, there was no significant difference in the liver oxidative stress parameters between infected-untreated mice (IC) and uninfected ones (HC), except the MDA concentration, where a 79.08% increase ($p < 0.01$) was recorded (Fig 4). In the PZQ3 group, we noted a decrease of 57.29% compared to the IC group. In contrast, the MDA

                    

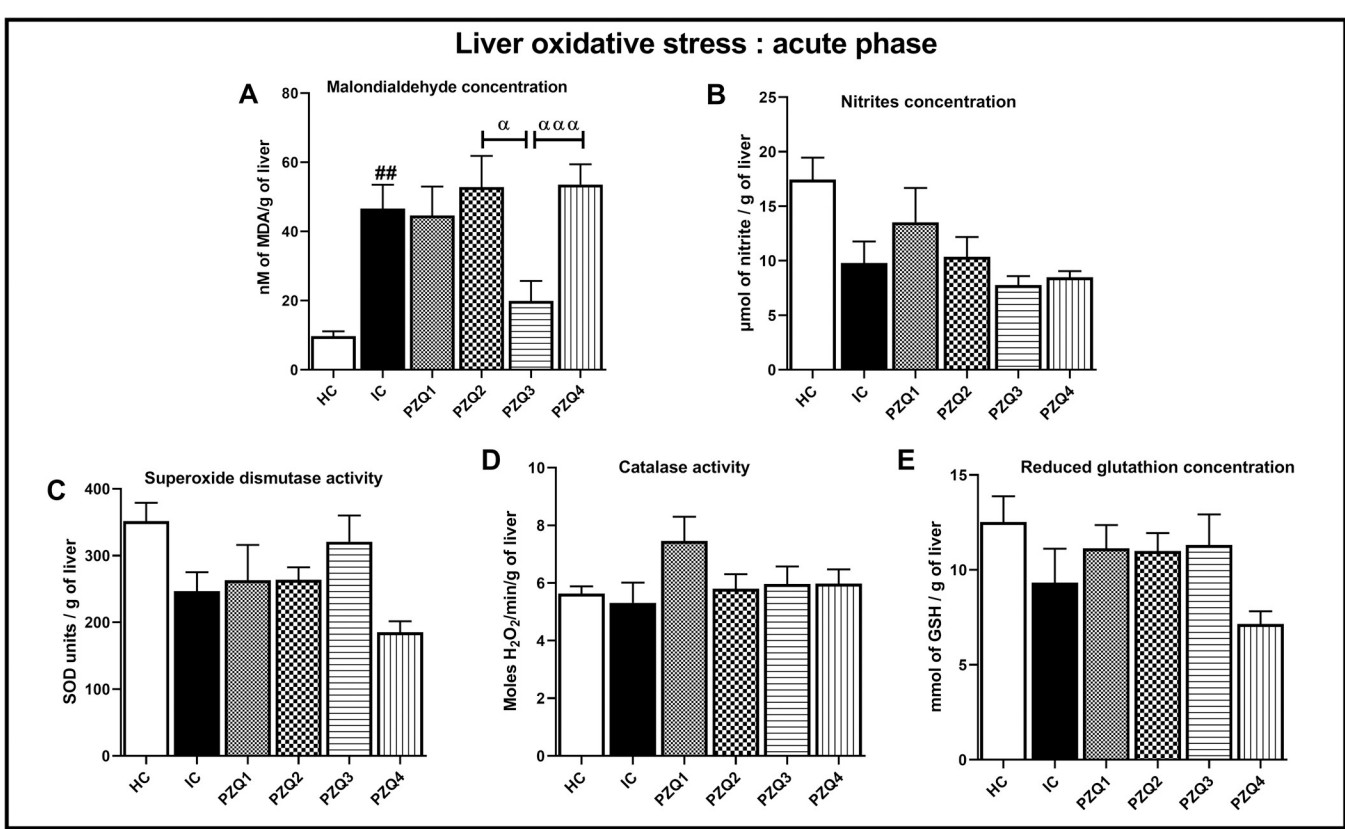

**Fig 4. Effect of praziquantel treatment on the liver oxidative status during the acute phase in *Schistosoma mansoni*-infected mice.** (A) malondialdehyde, (B) nitrites, (C) superoxide dismutase, (D) catalase, and (E) Reduced glutathione. HC: Uninfected mice, IC: Infected-untreated mice, PZQ1: Infected mice treated with praziquantel at 100 mg/kg for 5 consecutive days, PZQ2: Infected mice treated with praziquantel at 100 mg/kg for 28 consecutive days, PZQ3: Infected mice treated with praziquantel at 18 mg/kg for 28 consecutive days, PZQ4: Infected mice treated with a single dose of praziquantel at 500 mg/kg. Data are expressed as mean ± SEM (n = 6). ANOVA followed by Tukey multiple comparison test was also used for statistical analysis. ##p < 0.01: values are significantly different from those uninfected mice (group HC). αp < 0.05, αααp < 0.001: values are significantly different from those of infected mice treated with PZQ3 posology (group PZQ3).

concentration drastically increased in PZQ2 and PZQ4 groups and was 5-fold higher than in the HC group ($p < 0.01$).

In the granulomatous stage of the infection (Fig 5), the hepatic MDA concentration was 16-fold higher than that of uninfected mice ($p < 0.001$). In contrast, nitrite level, SOD and catalase activities were decreased by 78.21% ($p < 0.001$), 50.48% ($p < 0.01$) and 49.36% ($p < 0.01$), respectively. Among the four PZQ treatment schemes, only the PZQ3 regimen significantly improved the levels of these biomarkers. We observed a decrease of MDA concentration by 71.98% and an increase of nitrite level, SOD, and catalase activities by 70.23%, 46.75%, and 42.33%, respectively, compared to the IC group. Regarding the GSH concentration, we recorded no significant variation between groups.

In the spleen, the acute infection produced no change in the oxidative status, except significantly decreased nitrite concentrations ($p < 0.05$). PZQ3 treatment restored the nitrite level, increasing it by 68.05% ($p < 0.01$) in mice of this group, compared to the IC group (Fig 6).

In the granulomatous stage (Fig 7), *S. mansoni* infection led to an 8-fold higher splenic MDA concentration compared to uninfected mice ($p < 0.001$). Moreover, the level of nitrites was significantly reduced by 70.06% ($p < 0.001$). Concomitantly, the antioxidant capacity of infected-untreated mice (IC) was altered as significant reductions ($p < 0.001$) of SOD, catalase, and GSH

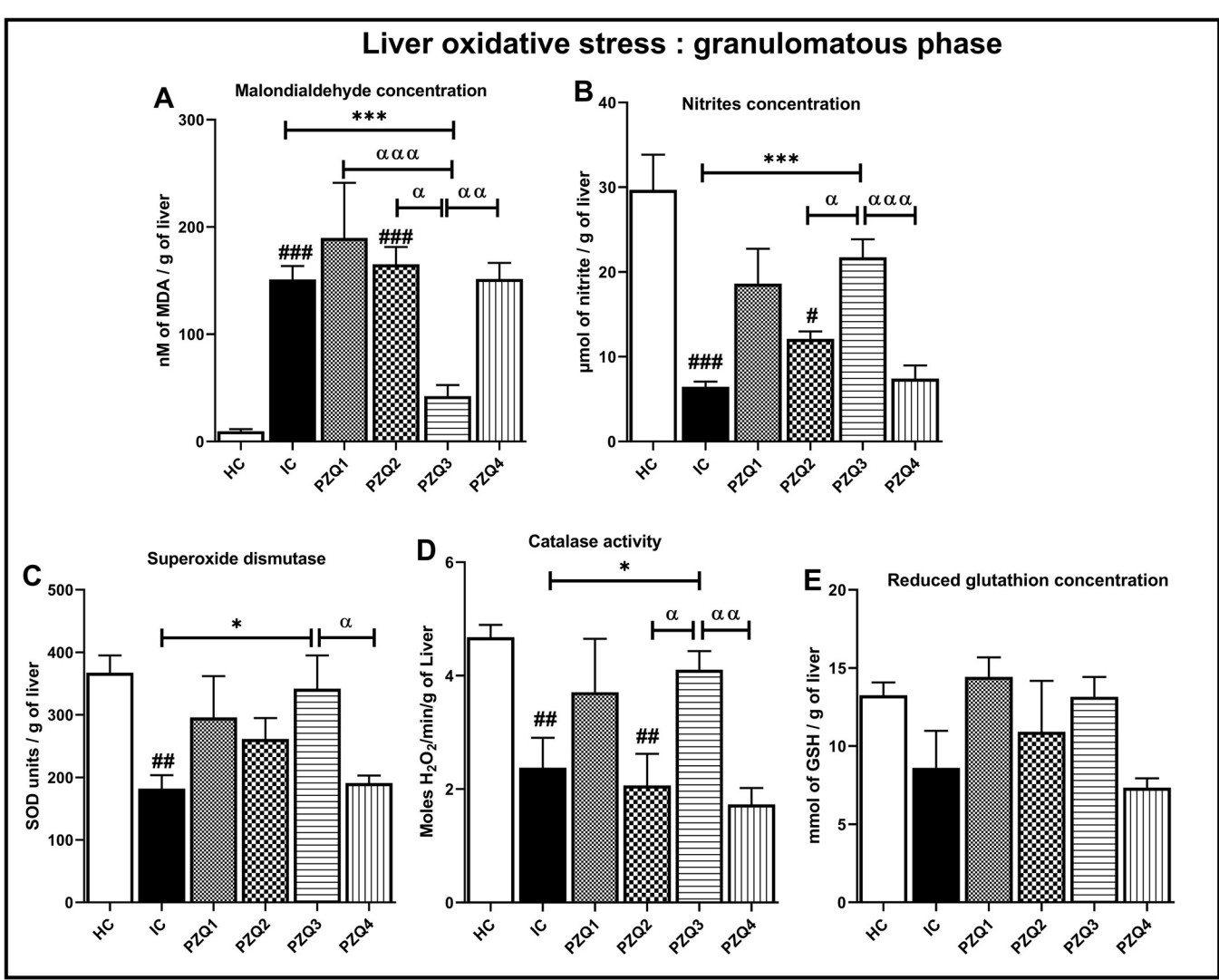

**Fig 5. Effect of praziquantel treatment on the liver oxidative status during the granulomatous phase in *Schistosoma mansoni*-infected mice.** (A) malondialdehyde, (B) nitrites, (C) superoxide dismutase, (D) catalase, and (E) Reduced glutathione. HC: Uninfected mice, IC: Infected-untreated mice, PZQ1: Infected mice treated with praziquantel at 100 mg/kg for 5 consecutive days, PZQ2: Infected mice treated with praziquantel at 100 mg/kg for 28 consecutive days, PZQ3: Infected mice treated with praziquantel at 18 mg/kg for 28 consecutive days, PZQ4: Infected mice treated with a single dose of praziquantel at 500 mg/kg. Data are expressed as mean ± SEM (n = 6). ANOVA followed by Tukey multiple comparison test was also used for statistical analysis. #p < 0.05, ###p < 0.001: values are significantly different from those uninfected mice (group HC). *p < 0.05, ***p < 0.001: values are significantly different from those of infected-untreated mice (group IC). $^{\alpha}$p < 0.05, $^{\alpha\alpha}$p < 0.01, $^{\alpha\alpha\alpha}$p < 0.001: values are significantly different from those of infected mice treated with PZQ3 posology (group PZQ3).

levels by 39.33%, 69.51%, and 77.34%, respectively, were recorded. The oxidative status significantly improved after treatment of mice with PZQ at 18 mg/kg/day for 28 consecutive days (PZQ3). A significant decrease of MDA concentration by 47.67% ($p < 0.05$), as well as an increase of nitrites concentration by 56.12% ($p < 0.01$) compared to infected-untreated mice, was recorded. Moreover, PZQ3 posology suppressed the *S. mansoni*-induced depletion of antioxidant capacity in the spleen by significantly increasing the SOD (25.18%) and catalase (53.27%) activities as well as GSH concentration (63.49%), compared to the IC group of mice (Fig 7).

Taken together, these results of the hepatic and splenic oxidative status parameters demonstrate a superior efficiency of the PZQ3 treatment regimen over the others (Figs 4, 5, 6 and 7).

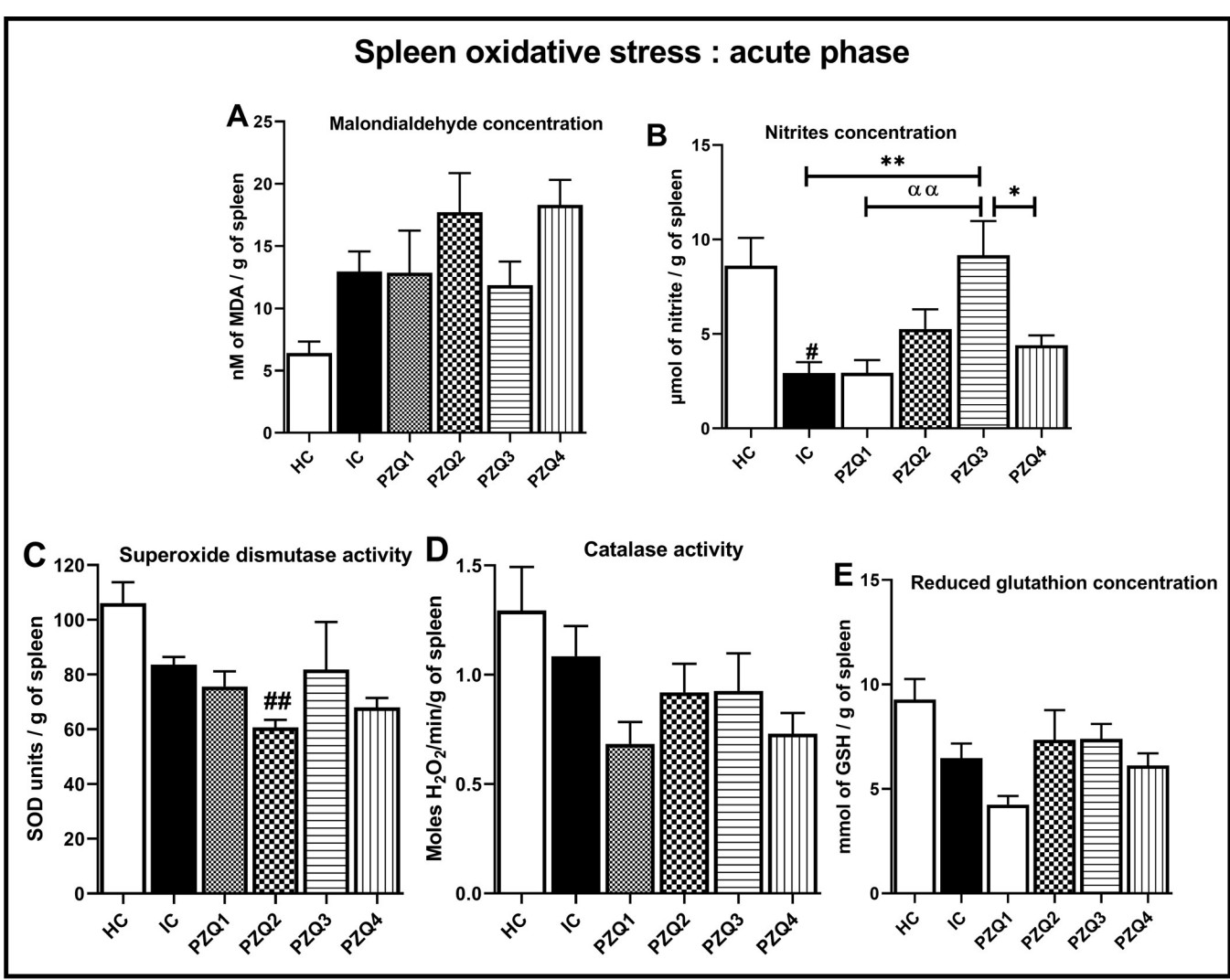

**Fig 6. Effect of praziquantel treatment on the spleen oxidative status during the acute phase in *Schistosoma mansoni*-infected mice.** (A) malondialdehyde, (B) nitrites, (C) superoxide dismutase, (D) catalase, and (E) Reduced glutathione. HC: Uninfected mice, IC: Infected-untreated mice, PZQ1: Infected mice treated with praziquantel at 100 mg/kg for 5 consecutive days, PZQ2: Infected mice treated with praziquantel at 100 mg/kg for 28 consecutive days, PZQ3: Infected mice treated with praziquantel at 18 mg/kg for 28 consecutive days, PZQ4: Infected mice treated with a single dose of praziquantel at 500 mg/kg. Data are expressed as mean ± SEM (n = 6). ANOVA followed by Tukey multiple comparison test was also used for statistical analysis. ##$p < 0.01$: values are significantly different from those uninfected mice (group HC). *$p < 0.05$, **$p < 0.01$: values are significantly different from those of infected-untreated mice (group IC). αα$p < 0.01$: values are significantly different from those of infected mice treated with PZQ3 posology (group PZQ3).

## Praziquantel treatment lessens *Schistosoma mansoni*-induced histopathological changes in the mouse liver and intestine

Histological examination of liver and ileum sections of uninfected mice showed typical structures (Fig 8I and 8II). Eight weeks after *S. mansoni* infection, granulomas were abundantly present around entrapped eggs in the liver and the mucosa and sub-mucosa of the ileum of IC mice. These granulomas were composed of central ova surrounded by laminated collagen fibers associated with inflammatory cells at the periphery. Compared to uninfected mice, we noticed a thickening of the *muscularis propria* of the ileum in all IC mice. Among PZQ-treated mice, the liver and ileum sections of PZQ1 and PZQ4 groups were similar to those of IC mice. In contrast, the liver of infected mice treated with either PZQ2 or PZQ3 regimens contained

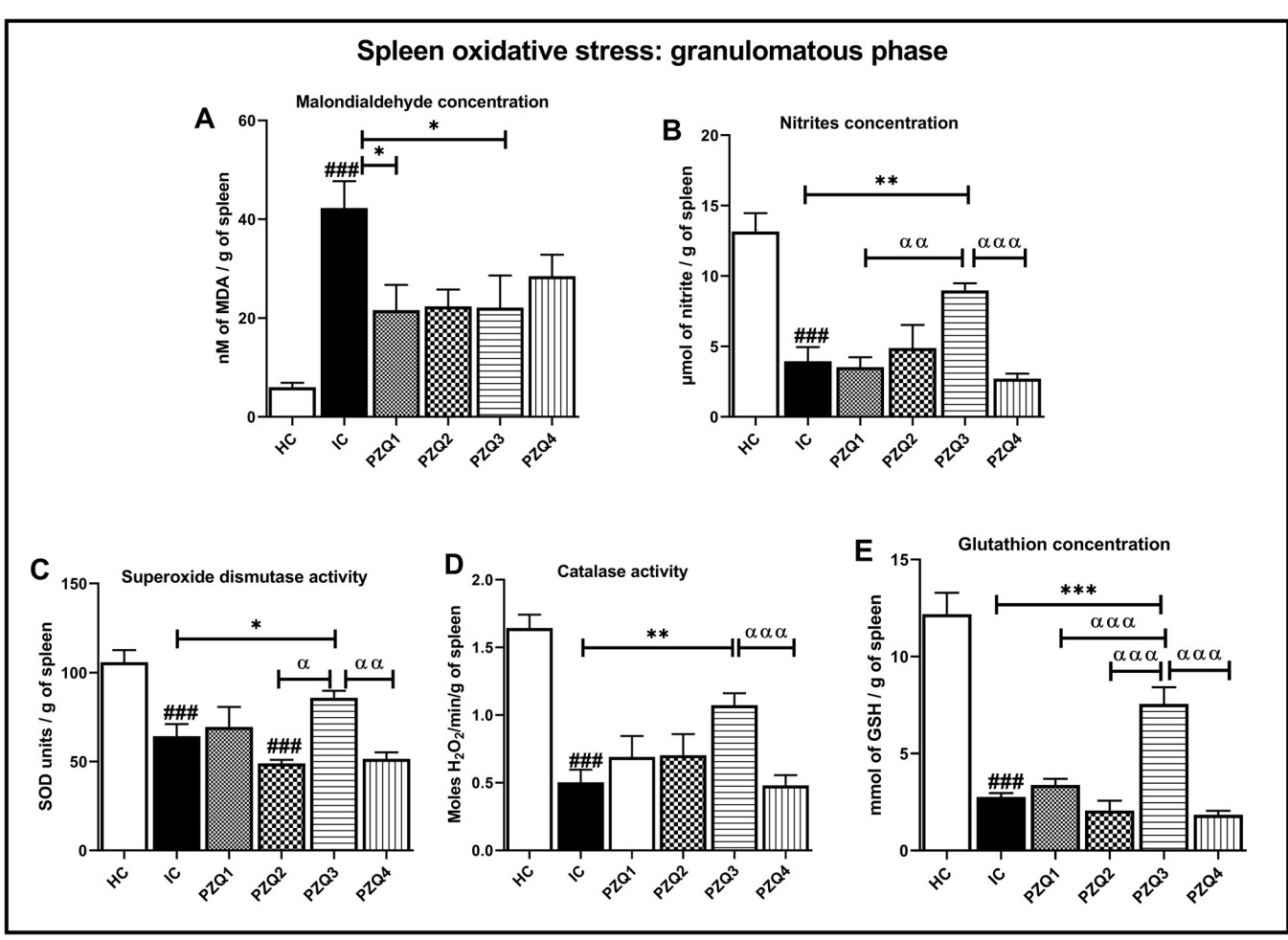

**Fig 7. Effect of praziquantel treatment on the splenic oxidative status during the granulomatous phase in *Schistosoma mansoni*-infected mice.** (A) malondialdehyde, (B) nitrites, (C) superoxide dismutase, (D) catalase, and (E) Reduced glutathione. HC: Uninfected mice, IC: Infected-untreated mice, PZQ1: Infected mice treated with praziquantel at 100 mg/kg for 5 consecutive days, PZQ2: Infected mice treated with praziquantel at 100 mg/kg for 28 consecutive days, PZQ3: Infected mice treated with praziquantel at 18 mg/kg for 28 consecutive days, PZQ4: Infected mice treated with a single dose of praziquantel at 500 mg/kg. Data are expressed as mean ± SEM (n = 6). ANOVA followed by Tukey multiple comparison test was also used for statistical analysis. $^{###}p < 0.001$: values are significantly different from those uninfected mice (group HC). $^{*}p < 0.05$, $^{**}p < 0.01$, $^{***}p < 0.001$: values are significantly different from those of infected-untreated mice (group IC). $^{\alpha}p < 0.05$, $^{\alpha\alpha}p < 0.01$, $^{\alpha\alpha\alpha}p < 0.001$: values are significantly different from those of infected mice treated with PZQ3 posology (group PZQ3).

few and small granulomas. Some of them were scarred. The ileum sections were nearly free of granuloma, fibrotic tissue, and few inflammatory cells were detected.

Morphometric analysis showed that the granulomas number in the liver and the ileum of infected-untreated mice (IC) was 25.34 ± 4.77 and 21.18 ± 3.09, respectively (Fig 9A and 9B). PZQ3 regimen significantly reduced the granuloma number in the liver by 93.21% (1.72 ± 0.60 vs. 25.34 ± 4.77) and in the ileum by 54.25% (9.69 ± 2.09 vs. 21.18 ± 3.09). The PZQ2 regimen also effectively reduced the number of granulomas by 90.68% and 80.35% in the liver and ileum, respectively. The number of hepatic and ileal granulomas in PZQ1 and PZQ4 treated mice was nearly similar to that in IC mice. Moreover, the granuloma volume regressed by 87.10% ($p < 0.001$) in the liver of PZQ3 mice (Fig 9C). PZQ2 treatment also decreased the hepatic granuloma volume (72.68%, $p < 0.001$), while the other regimens had no significant

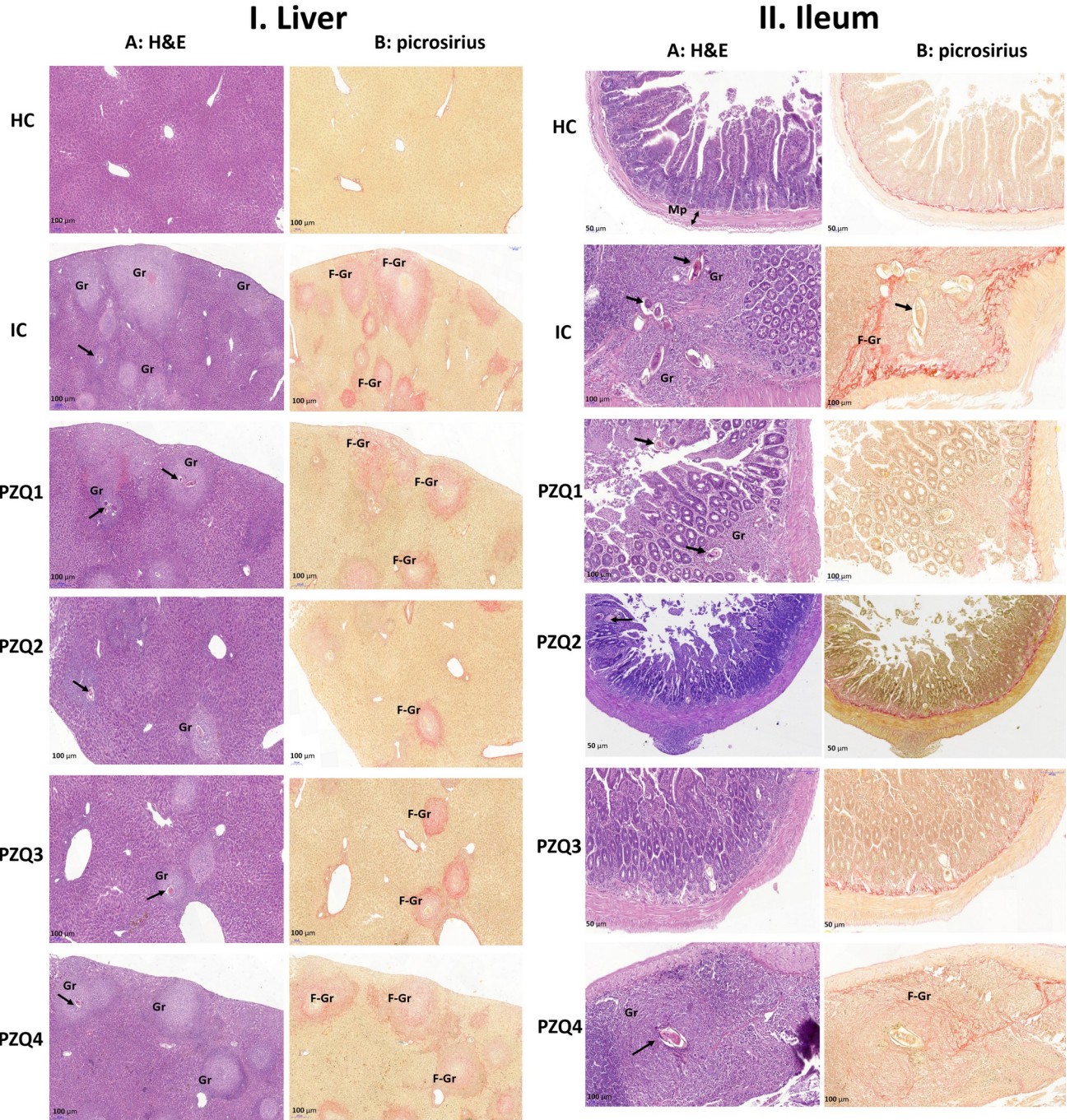

**Fig 8. Effect of praziquantel treatment on the liver (I) and the ileum (II) histology of *Schistosoma mansoni*-infected mice stained with hematoxylin and eosin (A) and picrosirius (B).** HC: Uninfected mice, IC: Infected-untreated mice, PZQ1: Infected mice treated with praziquantel at 100 mg/kg for 5 consecutive days, PZQ2: Infected mice treated with praziquantel at 100 mg/kg for 28 consecutive days, PZQ3: Infected mice treated with praziquantel at 18 mg/kg for 28 consecutive days, PZQ4: Infected mice treated with a single dose of praziquantel at 500 mg/kg. Simple arrows indicate *Schistosoma mansoni* egg; F-Gr: fibrotic granuloma; Gr: granuloma; bidirectional arrow indicates the height of the *muscularis propria* (Mp).

effect. Accordingly, the ileal granuloma volume (Fig 9D) was most markedly reduced after the PZQ3 regimen (89.59% reduction, $p < 0.001$). PZQ1 and PZQ2 regimens reduced ileal granuloma volumes by 75.59 and 81.59% ($p < 0.05$), respectively, while no significant difference was

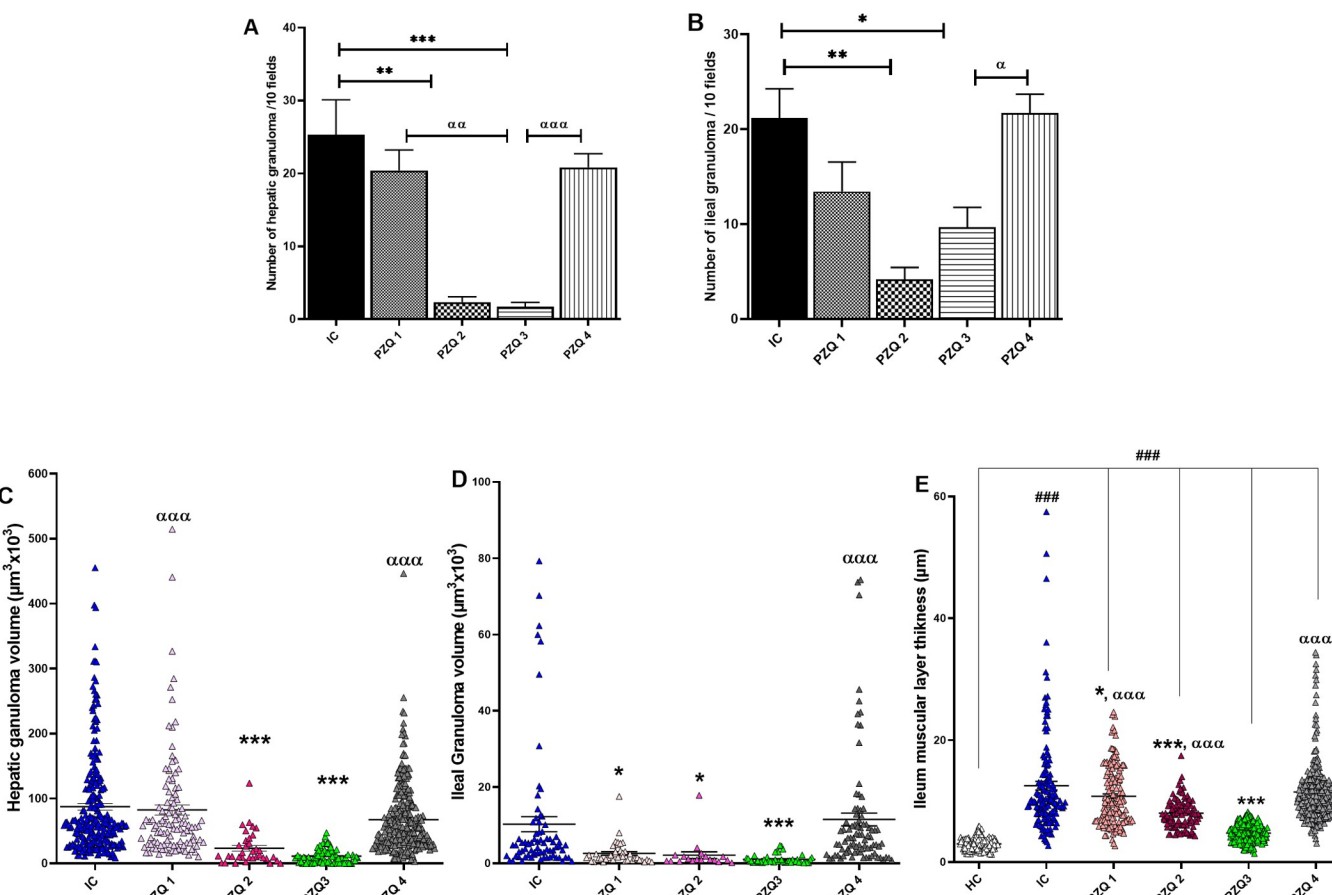

**Fig 9. Effect of praziquantel treatment on the morphometry of the liver and the ileum of *Schistosoma mansoni*-infected mice. (A)** number of liver's granulomas; **(B)** number of ileal granulomas; **(C)** volume of hepatic granulomas; **(D)** volume of ileal granulomas; **(E)** muscular layer thickness of the ileum. HC: Uninfected mice, IC: Infected-untreated mice, PZQ1: Infected mice treated with praziquantel at 100 mg/kg for 5 consecutive days, PZQ2: Infected mice treated with praziquantel at 100 mg/kg for 28 consecutive days, PZQ3: Infected mice treated with praziquantel at 18 mg/kg for 28 consecutive days, PZQ4: Infected mice treated with a single dose of praziquantel at 500 mg/kg. Data are expressed as mean ± SEM (n = 6). ANOVA followed by Tukey multiple comparison test was also used for statistical analysis. ###$p < 0.001$: values are significantly different from those uninfected mice (group HC). *$p < 0.05$, **$p < 0.01$, ***$p < 0.001$: values are significantly different from those of infected-untreated mice (group IC). α$p < 0.05$, αα$p < 0.01$, ααα$p < 0.001$: values are significantly different from those of infected mice treated with PZQ3 posology (group PZQ3).

observed in the PZQ4 group. The thickness of the muscular layer of the ileum was also measured (Fig 9E). In both IC and PZQ-treated groups, the ileal muscular layer was significantly enlarged ($p < 0.001$) compared to the HC group. PZQ1, PZQ2 and PZQ3 regimens successfully reduced the thickness of the muscular layer by 13.85% ($p < 0.05$), 35.98% ($p < 0.001$) and 61.86% ($p < 0.001$), respectively, compared to that of IC mice. Due to the absence of granuloma at the acute stage of the infection, histopathological analysis and morphometry were not performed on the liver and the ileum of mice.

### Protection induced by praziquantel early treatment leads to the modulation of the immune response in *Schistosoma mansoni*-infected mice

The level of selected Th-1, Th-2, and Th-17 cytokines was measured in mice's sera after 5 (acute stage) and 8 weeks (granulomatous stage) *p.i.*

At 5 weeks *p.i.*, infected-untreated mice showed significant high levels of IFN-γ ($p < 0.05$), TNF-α ($p < 0.01$), IL-5 ($p < 0.05$) and IL-17 ($p < 0.05$), while the level of the regulatory

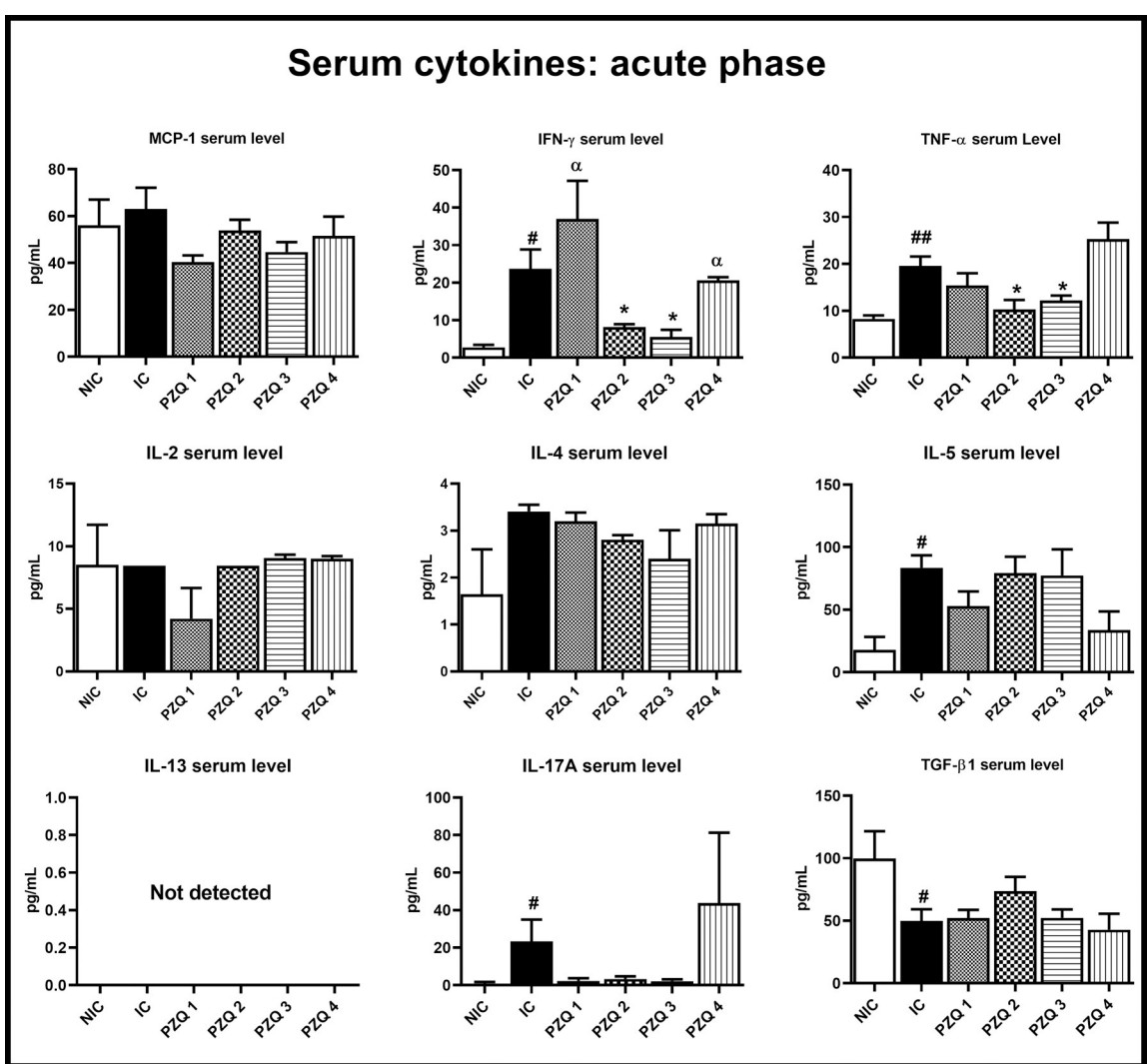

**Fig 10. Effect of praziquantel treatment on the serum cytokine levels during the acute stage of *Schistosoma mansoni* infection in mice.** HC: Uninfected mice, IC: Infected-untreated mice, PZQ1: Infected mice treated with praziquantel at 100 mg/kg for 5 consecutive days, PZQ2: Infected mice treated with praziquantel at 100 mg/kg for 28 consecutive days, PZQ3: Infected mice treated with praziquantel at 18 mg/kg for 28 consecutive days, PZQ4: Infected mice treated with a single dose of praziquantel at 500 mg/kg. Data are expressed as mean ± SEM (n = 6). ANOVA followed by Tukey multiple comparison test was also used for statistical analysis. #p < 0.05, ##p < 0.01: values are significantly different from those uninfected mice (group HC). *p < 0.05: values are significantly different from those of infected-untreated mice (group IC). αp < 0.05: values are significantly different from those of infected mice treated with PZQ3 posology (group PZQ3).

cytokine TGF-β was diminished (*p* < 0.05), compared to HC mice. The MCP-1 and IL-4 were increased by 11.04% and 51.76% in IC mice, but these differences were not statistically significant. Among PZQ-treated groups at this infectious stage, there was a significant reduction (*p* < 0.05) of IFN-γ and TNF-α levels in the PZQ2 and PZQ3 groups compared to those of the IC group. IL-17 level was also markedly reduced. No differences were observed in PZQ1 and PZQ4-treated mice compared to IC mice. Interestingly, no IL-13 was detected in the serum of any mouse at this stage (Fig 10).

At 8 weeks *p.i.*, MCP-1, IFN- γ, TNF-α, IL-4, IL-5, IL-13, and IL-17 levels were significantly (*p* < 0.01) increased in IC mice, compared to HC mice, while the TGF-β level was significantly (*p* < 0.01) decreased. Regarding the treatment groups, PZQ3 regimen was the most efficient as

it reduces the MCP-1 ($p < 0.01$), Th-1 ($p < 0.05$), Th-2 ($p < 0.01$) and Th17 cytokines ($p < 0.05$) levels and increases the level of TGF-β ($p < 0.01$). Regarding the other PZQ treatment regimens, PZQ1 and PZQ4 modified the levels of IL-17 and TGF-β; but also of IL-13 for PZQ1. Except for the IL-5 level, no statistical difference was recorded between PZQ2 and IC groups (Fig 11).

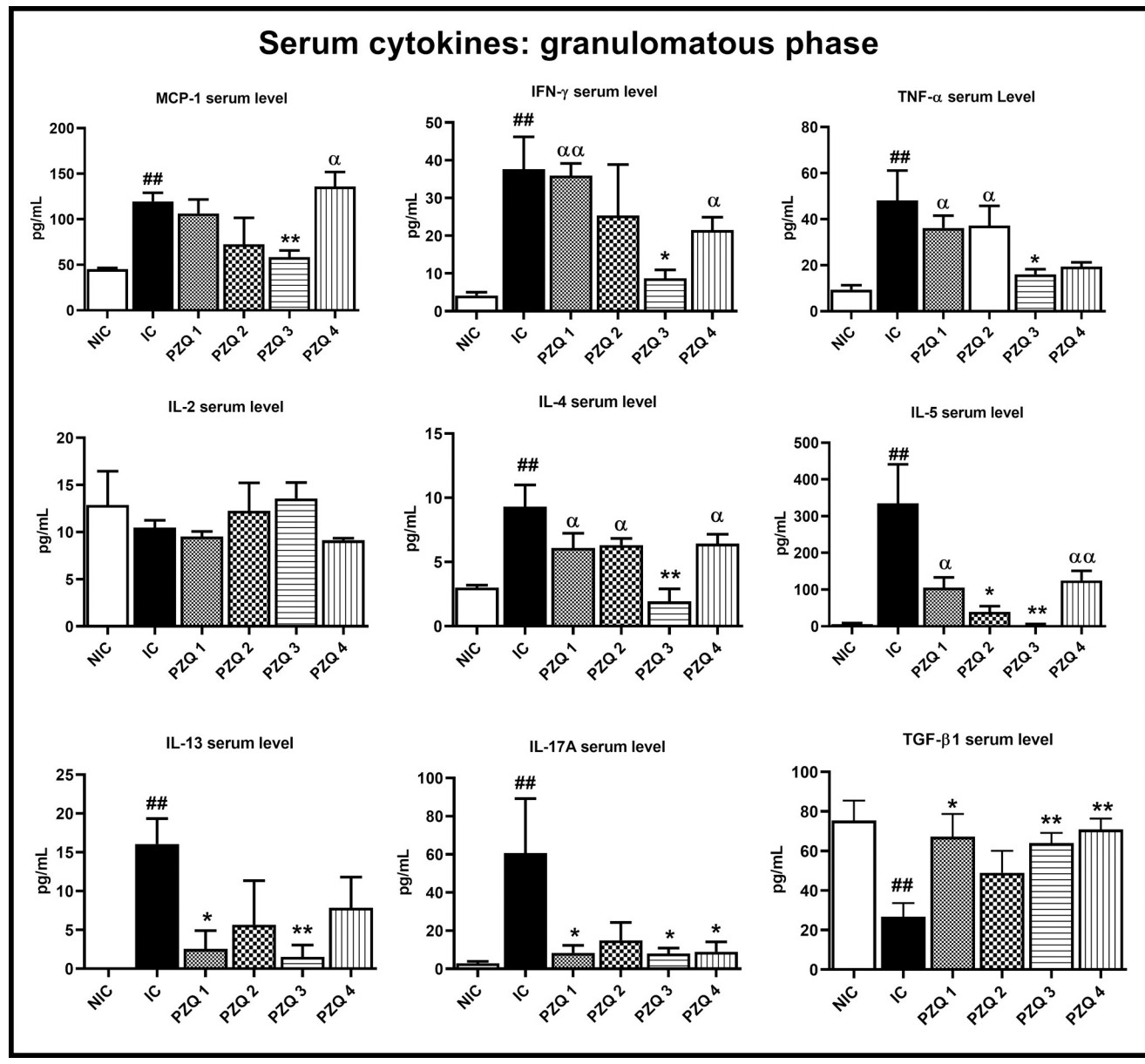

**Fig 11. Effect of praziquantel treatment on the serum cytokines levels during the granulomatous stage of *Schistosoma mansoni* infection in mice.** HC: Uninfected mice, IC: Infected-untreated mice, PZQ1: Infected mice treated with praziquantel at 100 mg/kg for 5 consecutive days, PZQ2: Infected mice treated with praziquantel at 100 mg/kg for 28 consecutive days, PZQ3: Infected mice treated with praziquantel at 18 mg/kg for 28 consecutive days, PZQ4: Infected mice treated with a single dose of praziquantel at 500 mg/kg. Data are expressed as mean ± SEM (n = 6). ANOVA followed by Tukey multiple comparison test was also used for statistical analysis. ##$p < 0.01$: values are significantly different from those uninfected mice (group HC). *$p < 0.05$, **$p < 0.001$: values are significantly different from those of infected-untreated mice (group IC). α$p < 0.05$, αα$p < 0.01$: values are significantly different from those of infected mice treated with PZQ3 posology (group PZQ3).

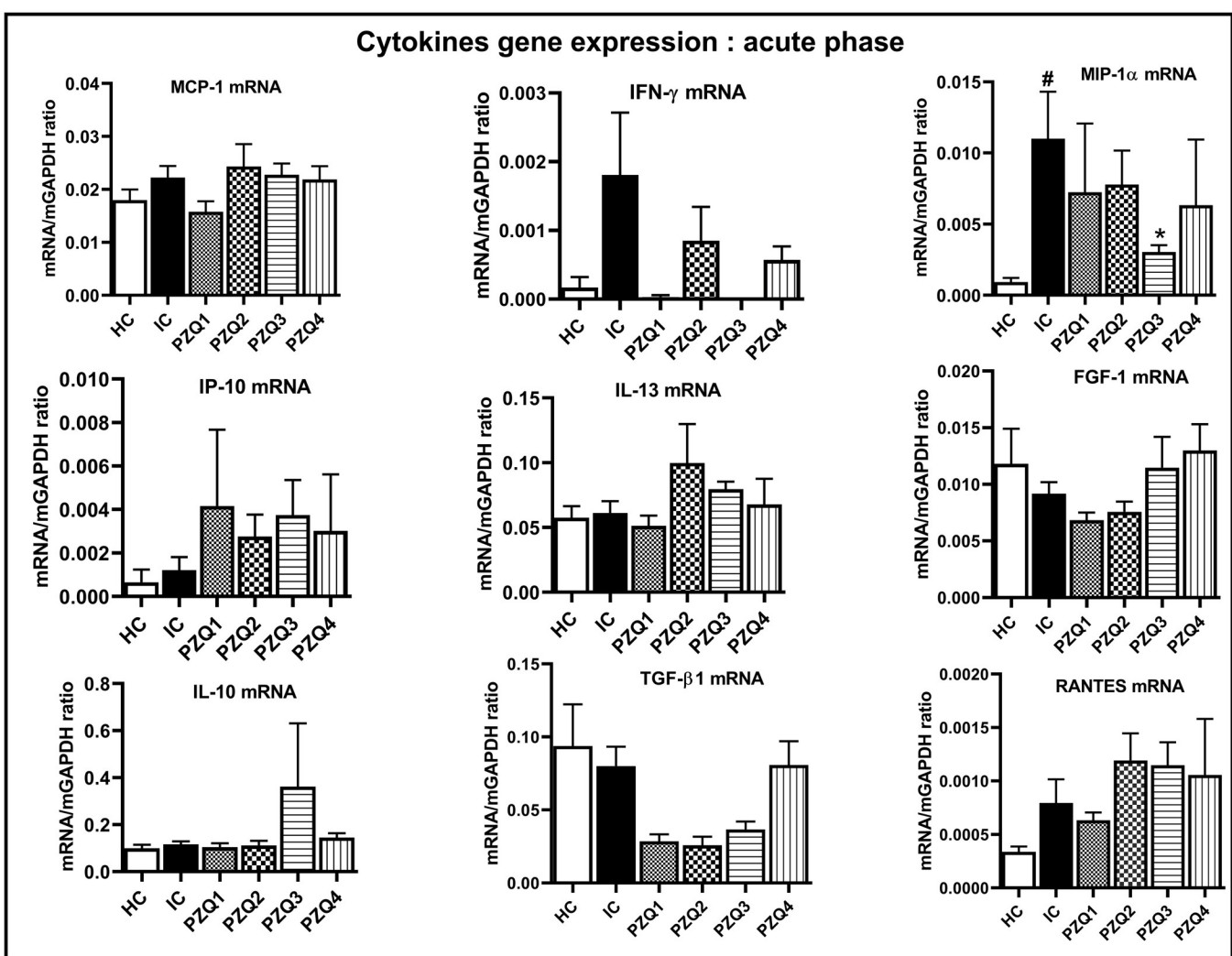

**Fig 12. Effect of praziquantel treatment on the liver mRNA expression of cytokines and chemokines during the acute stage of *Schistosoma mansoni* infection in mice.** HC: Uninfected mice, IC: Infected-untreated mice, PZQ1: Infected mice treated with praziquantel at 100 mg/kg for 5 consecutive days, PZQ2: Infected mice treated with praziquantel at 100 mg/kg for 28 consecutive days, PZQ3: Infected mice treated with praziquantel at 18 mg/kg for 28 consecutive days, PZQ4: Infected mice treated with a single dose of praziquantel at 500 mg/kg. Data are expressed as mean ± SEM (n = 6). ANOVA followed by Tukey multiple comparison test was also used for statistical analysis. [#]$p < 0.05$: values are significantly different from those uninfected mice (group HC). [*]$p < 0.05$: values are significantly different from those of infected-untreated mice (group IC).

The gene expression of cytokines and chemokines involved in the immune response against schistosomiasis was evaluated in infected mice's livers.

After 5 weeks of infection, the only significant difference was observed for MIP-1α, where infection led to a marked increase in mRNA levels, essentially reversed by PZQ3 treatment. Of note, the gene expression of IFN-γ increased by 90.60% in the liver of infected mice (IC) compared to that of uninfected mice (HC) and was nearly absent in PZQ3 treated mice (Fig 12).

At the granulomatous phase of the infection (8 weeks *p.i*), *S. mansoni* induced a significant increase of gene expression of IFN-γ ($p < 0.05$), MIP-1α ($p < 0.01$), and MCP-1 ($p < 0.01$) in the liver of IC mice, in comparison to HC group. An increase by 99.56% of the expression of IP-10 in IC mice was noted despite the no statistical significance. The profibrotic genes (IL-13 and FGF-1) were also significantly expressed, while the regulatory genes (IL-10, TGF-β, and RANTES) were unexpressed in the liver of infected mice. In contrast, FGF-1 and IL-13 mRNA

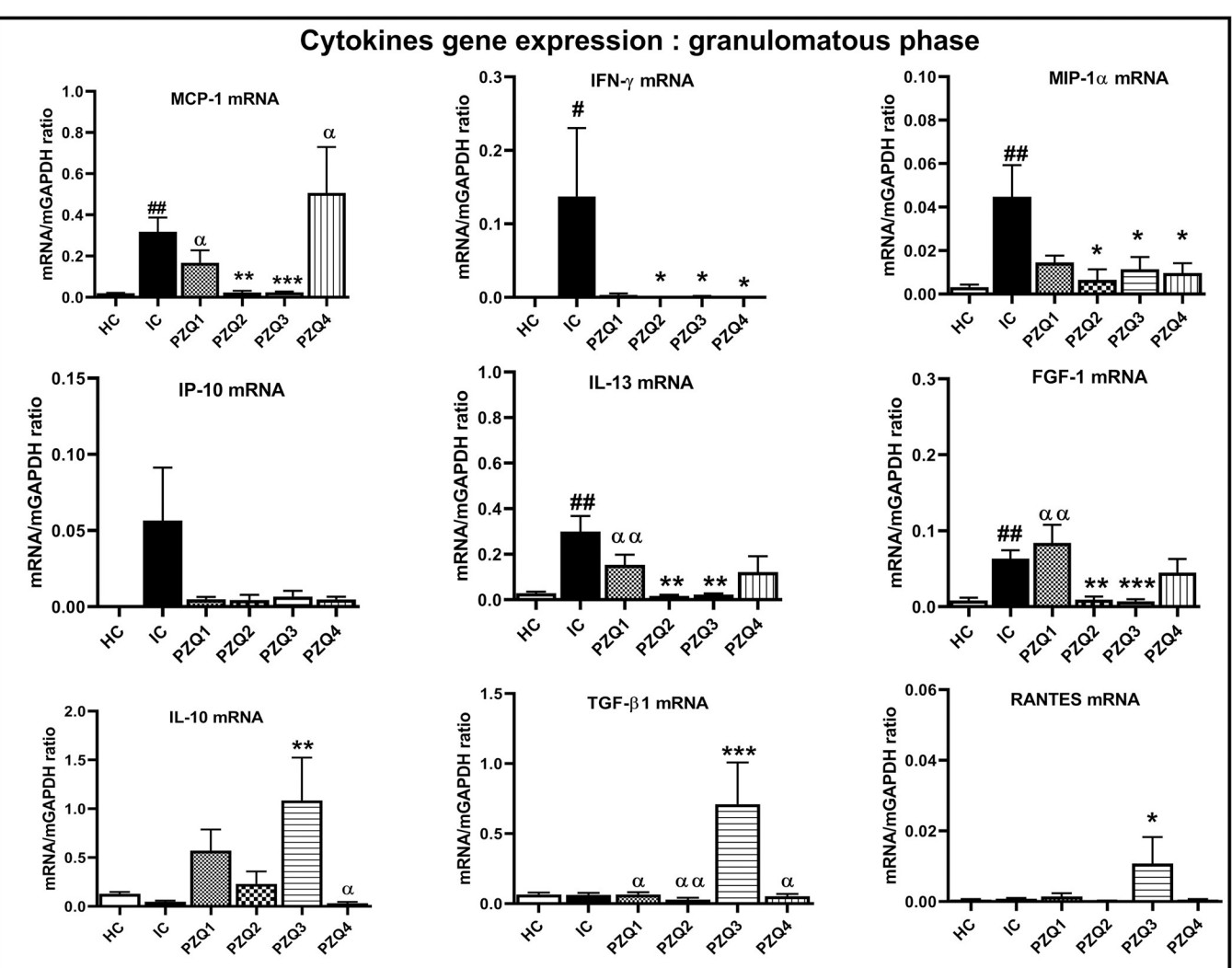

**Fig 13. Effect of praziquantel treatment on the liver mRNA expression of cytokines and chemokines during the granulomatous stage of *Schistosoma mansoni* infection in mice.** HC: Uninfected mice, IC: Infected-untreated mice, PZQ1: Infected mice treated with praziquantel at 100 mg/kg for 5 consecutive days, PZQ2: Infected mice treated with praziquantel at 100 mg/kg for 28 consecutive days, PZQ3: Infected mice treated with praziquantel at 18 mg/kg for 28 consecutive days, PZQ4: Infected mice treated with a single dose of praziquantel at 500 mg/kg. Data are expressed as mean ± SEM (n = 6). ANOVA followed by Tukey multiple comparison test was also used for statistical analysis. $^{\#}p < 0.05$, $^{\#\#}p < 0.01$: values are significantly different from those uninfected mice (group HC). $^{*}p < 0.05$, $^{**}p < 0.001$, $^{***}p < 0.001$: values are significantly different from those of infected-untreated mice (group IC). $^{\alpha}p < 0.05$, $^{\alpha\alpha}p < 0.01$: values are significantly different from those of infected mice treated with PZQ3 posology (group PZQ3).

levels were significantly higher ($p < 0.01$) by 87.06% and 90.67%, respectively, in infected mice's liver than in uninfected ones. In PZQ2, PZQ3, and PZQ4 regimens, the gene expression of INF-γ, IP-10, MIP-1α, IL-13, and FGF-1 was restored to near-normal levels. Interestingly, immune regulators genes were more expressed in PZQ3-treated mice than IC mice. A significant increase in the gene expression of IL-10 (95.57%), RANTES (92.58%), and TGF-β1 (91.11%) was observed in the liver of PZQ3 mice in comparison to that of IC mice (Fig 13).

Considering these results, infection led to a considerable increase of most immune factors, more markedly so in the granulomatous stage, especially for Th2 cytokines. Notably, the regulatory cytokines were less visible in infected mice, indicating a pathological inflammatory reaction. Among the treatment regimens, PQZ3 was the most efficient in reversing this unbalanced inflammatory reaction.

## Discussion

*Schistosoma mansoni* infection is relatively insidious until eggs released by females and destined to be excreted via the intestine become trapped in the liver and gastrointestinal tract tissues. Eggs induce potent inflammatory responses in these organs, leading to severe granulomatous inflammation and fibrosis. The pathology is associated with anemia, abdominal pain, diarrhea, and growth and cognitive impairments [31,32]. PZQ is the only drug readily available for the treatment of schistosomiasis. Because of its low cost, safety, and efficacy, mass PZQ administration has been the key intervention for schistosomiasis control programs [1,18]. One of the main challenges in schistosomiasis treatment is to avoid eggs production in the vertebrate host by administrating the appropriate PZQ posology at the initial stage of the infection. In the present study, we found that, depending on the regimen administered, PZQ can be effective during the migratory phase of *S. mansoni*. Our results show that PZQ kills immature forms of schistosomes, *in vivo*, after an oral administration of 18 mg/kg/day or 100 mg/kg/day for 28 consecutive days to mice. Interestingly, the herein reported PZQ regimens promoted a significant and high decrease of hepatic, intestinal, and fecal eggs load. These results attest that these two treatment regimens could considerably decrease the worm fecundity of surviving worms. Similar results had been obtained by Hoekstra *et al.*[33] in *S. mansoni*-infected children after treatment with four repeated doses of 40mg/kg of PZQ at two weeks intervals for six weeks.

Several authors have reported a body loss or low body weight gain associated with hepatosplenomegaly and high intestine index during experimental schistosomiasis mansoni in mice [19,34,35]. Results from our study revealed that *S. mansoni*-infected mice exhibited significant body weight loss, hepatosplenomegaly, and high intestine and thymus indices after 56 days of infection. Enlargement of the liver and the intestine could be due to *S. mansoni* eggs deposition in these tissues. Spleen and thymus weight gain could result from a high activation of these organs during the immune response against the parasite. In this study, oral administration of PZQ at 18mg/kg/day for 28 consecutive days to *S. mansoni*-infected mice prevents weight loss and hepatosplenomegaly and the high intestine thymus indices. The reduction of the liver, spleen, intestine, and thymus indices could be the consequence of the reducing number of eggs trapped in organs consecutively to the schistosomicidal activity of PZQ at these two treatments schemes.

*S. mansoni* infection is classically responsible for multi-cellular granulomatous inflammation surrounding eggs trapped in various tissues [3]. *S. mansoni* infection is responsible for developing fibrosis around eggs granuloma within the portal tract. This fibrosis is a complex process subsequent to increased synthesis and deposition of extracellular matrix components [3]. *S. mansoni*-induced pathology results from egg embolization in host tissues, particularly the liver and intestine. Therefore, granuloma primarily forms around those eggs lodged in the presinusoidal capillary venules of the liver. These granulomas sometimes give rise to severe fibrosis, disrupting blood flow through the liver, causing portal hypertension and portocaval shunting. The intestinal form of the disease is characterized by a mucosal granulomatous response that leads to pseudo polyposis, micro-ulceration, and superficial bleeding [3,36]. In tissues, parasite egg antigens induce granulomatous inflammatory reactions involving macrophages, eosinophils, lymphocytes, and hepatic stellate cells. Activated hepatic stellate cells synthesize collagen I and III and alpha-smooth muscle, contributing to the scar tissue constituting liver fibrosis [37–39]. Pearce and McDonald [2] stated that granuloma formation requires the recruitment of inflammatory cells from the bloodstream via up-regulation of adhesion molecules on activated vascular endothelial cells, induced by cytokines and chemokines released at the site of inflammation. In the present work, early treatment with 18 mg/kg/day or 100 mg/

kg/day of PZQ for 28 days induced a noticeable degree of protection corresponding to less severe pathological changes in the liver and intestine of infected mice, particularly the reduction of the inflammatory reaction and the granuloma number and size. The significant reduction of the number and the size of fibrotic granulomas in the liver and the intestine and the reduction of the ileal muscular thickness was correlated to the decreased hepatic and intestinal egg loads consecutive the reduction of worm burden in PZQ treated mice [34,35]. Moreover, Nono et al. [40] demonstrated that PZQ prevents the onset of fibrosis process before it is established.

Schistosome eggs induce liver fibrosis, a typical pathological process that can lead to irreversible cirrhosis and the inability of the liver to perform its biochemical function [38,39]. Previous studies have shown that liver pathology during acute schistosomiasis in mice (5 to 6 weeks *p.i*) is associated with liver function impairment and oxidative stress disbalance and is exacerbated with the progression of the infection (8 to 12 weeks *p.i*) [41–43]. It has been demonstrated that in *S. mansoni*-infected mice, schistosome eggs deposition in the liver induces granulomatous reaction, leading to the release of transaminases from the injured hepatic cells to the bloodstream [19,29,35,44–46]. In agreement with the reports of Jatsa et al. [34,35], the increment of such enzymes in serum is probably due to the destruction of hepatocytes by the action of parasite eggs-releasing toxins. In addition, Fahmy et al. [46] reported that hepatocyte membrane damage seems to be the prime culprit for the marked increase in the serum marker enzymes, AST, ALT, and ALP following *Schistosoma* infection. In conjunction with the report of Fahmy et al. [46], data from the present study showed that treatment of infected mice with 18 mg/kg/day of PZQ for 28 days during the larval and juvenile stages of the parasite significantly decreases the serum activities of ALT, AST, ALP, and total bilirubin concentration and increased γ-GT activity. Moreover, the primary cause of the metabolic dysfunction in schistosomiasis is the site-specific oxidative damage of some of the susceptible amino acids of protein [47,48]. Following the studies of Jatsa et al. [19,34,35], the present study shows that *S. mansoni* infection in mice induces a significant decrease in the serum total proteins concentration. Similar to what Fahmy et al. [45] reported, data from the present study shows that treatment with PZQ at the early stage of infection significantly increases the concentration of serum total proteins in infected-treated groups.

The infection with *S. mansoni* is responsible for the early production of reactive oxygen species (ROS), inducing oxidative stress in the liver [19,34,35,44,49] and the spleen[50]. In the present work, *S. mansoni* impaired the antioxidant defense since the levels of some endogenous antioxidants were depleted. Moreover, MDA, a lipid peroxidation product, was overproduced. Lipid peroxidation has been recognized as a significant consequence of oxidative damage in schistosomiasis [43]. The marked depletion of GSH level, SOD and catalase activities, keys endogenous antioxidants, in infected untreated mice reflects hepatic and spleen cells damage. This depletion could be linked to the increased oxidative stress and the cytotoxicity activity of $H_2O_2$, which is produced due to inhibition of glutathione reductase activity [41]. Since PZQ administration at 18 mg/kg/day for 28 days increases the levels of GSH and MDA, as well as SOD and catalase activities, it appears that this treatment prevents oxidative stress and preserves the antioxidant capacity of the liver and spleen cells. This effect might be correlated to the ability of PZQ to kill immature worms and avoid eggs deposition in organs.

Several authors have reported anemia during acute and chronic stages of schistosomiasis. In fact, schistosomiasis is associated with a decrease in red blood cells (RBC) and hemoglobin levels, as well as hematocrit percentage [51,52]. Data obtained in the present work revealed significant decrements in the mean RBC count, hemoglobin level, and hematocrit percentage in infected mice after 56 days. Anemia induced by *S. mansoni* infection could be explained by the bleedings caused by *S. mansoni* ova extrusion through the blood vessels and the intestinal wall

[41] or by consuming RBC and hemoglobin by schistosome adult worms [7]. The schistosome's blood-feeding allows them to acquire amino acids to grow, develop, and reproduce [7]. Hemoglobin is then released into the cecum of schistosomes, where it is degraded in amino acids necessary for parasite proteins synthesis. Moreover, hemoglobin is likely the only protein of the host that can supply schistosomes with amino acids [7,51]. In our study, infected mice treated with PZQ improved RBC count, hemoglobin, and hematocrit contents. By reducing the parasite burden, treatment with 18 mg/kg/day or 100 mg/kg/day of PZQ for 28 days at the early stage of infection limited intestinal bleedings and schistosome feeding on host RBC. Our study also exhibited a significant increment in total leukocytes and granulocytes counts in *S. mansoni*-infected untreated mice compared to uninfected mice.

Interestingly, infected mice presented high peripheral blood granulocytes at 5 and 8 weeks of infection. Blood granulocytes are critical in host defense mechanisms against invading pathogens as they are rapidly recruited to the infection site [53]. In addition, these cell types are present at the initiation and maintenance of chronic allergic inflammation and could play a protective role in the immune response against *S. mansoni* [54–57]. Total leukocytes and granulocytes reached near-normal levels when infected mice were treated with 18 mg/kg/day of PZQ for 28 days. This restoration could be attributed to PZQ's ability to kill parasites and limit worms' immunogenic properties and intratissular eggs [48]. The thrombocytopenia recorded in *S. mansoni* infected-untreated mice in this study is in conjunction with several authors who demonstrated that a low blood platelet count is associated with *Schistosoma* infection in mice [58–60] or humans [61,62]. Stanley et al. [60] reported that thrombocytopenia occurs in *S. mansoni*-infected mice just a few days after penetration of the parasite larvae in the bloodstream. In addition, Petroianu et al. [63] demonstrated that thrombocytopenia is also associated with splenomegaly and liver fibrosis during the hepatosplenic stage of schistosomiasis. Two ways can explain this thrombocytopenia. First, schistosomes worms interfere with primary hemostasis by inhibiting the ADP-mediated platelets activation and aggregation. In fact, the schistosome tegument contains several enzymatic activities that could lead to the degradation of extracellular ATP or ADP. In this manner, the enzyme expressed at the tegumental surface helps limit blood clot formation around the worms by acting as an anticoagulant and permitting the parasites free movement within the vasculature [59,60,64]. Secondly, platelets are sequestrated in the spleen of infected mice, leading to splenomegaly observed in hepatosplenic schistosomiasis [63,65]. In the present study, the treatment of infected mice with PZQ at 18 mg/kg/day for 28 consecutive days prevented thrombocytopenia. This result could be the consequence of the low parasitic burden that limits the sequestration of platelets in the spleen, as confirm by the average spleen index.

Experimental models of hepatosplenic schistosomiasis are used to appreciate the involvement of a CD4+ T helper (Th) cell response in the progression of the disease. A Th1 response is initiated at the early acute stage of infection and is directed against the migrating schistosomula and immature adult worms [2,66,67]. This response is characterized by increased expression of the Th1 proinflammatory cytokines IL-1, IL-12, TNF-α, and IFN- γ. The immune response switches to a pronounced Th2 response at 6 to 8 weeks *p.i* following the onset of egg deposition. This Th2 response is characterized by the production of high levels of Th2 cytokines such as IL-4, IL-5, IL-10, and IL-13 [4,68]. As the infection progresses to a more chronic stage in the moderate or controlled disease, general down-modulation of the Th2 response is observed [4,67,69]. This immunomodulation might be driven by IL-10 and TGF-β, which induce T regulatory cells (Tregs) that regulate the balance of Th1/Th2 response [2,4]. At the acute stage of the infection, we noticed a predominantly Th1 immune response characterized by a high level of serum IFN- γ, TNF-α, and IL-17 and the increased gene expression of IFN-γ, MIP-1α and IP-10 in the liver of infected-untreated mice. In infected-untreated mice,

hyperpolarized Th2 and Th17 immune responses were observed at the granulomatous stage. They expressed higher serum levels of IL-4, IL-5, and IL-13 and the hepatic mRNA of IL-13, FGF-1, MCP-1, and MIP-1α than those of uninfected mice. In addition, IL-17 was hardly detected at any infectious stage and indicated a chronic inflammatory process associated with severe liver pathology [67,70,71].

Moreover, decreased serum levels of immunomodulators (IL-10 and TGF-β) and their liver gene expression associated with a low expression of the RANTES gene were recorded in infected-untreated mice. According to Burke et al. [4], during the first 4–6 weeks of infection in the mouse, a moderate Th1 response is generated against migrating schistosomula and immature adult worms. This response is characterized by increased circulating proinflammatory cytokines [4,67]. Elevated levels of these cytokines have also been associated with the development symptoms of acute infection in humans [2,72]. Th17 response has been linked with severe hepatic inflammation in schistosomiasis [67,71]. However, the exact role of Th17 cells in regulating granulomatous pathology remains unclear. Still, Chen et al. [73] reported the detection of Th17 cells in *S. japonicum*-infected liver, suggesting that the increased levels of Th17 cell populations are correlated with more severe pathology. These results corroborated our findings on IL-17 production with *S. mansoni*-infected mice. In fact, several authors have demonstrated that the exacerbation of granulomatous lesions in *S. mansoni*-infected mice was directed by a Th17 response [4,36,73,74]. In addition, IL-17 neutralization reduced granulomatous inflammation and the expression of proinflammatory cytokines and chemokines in *S. mansoni*-infected mice [67,75]. Similarly, the highest level of MIP1-α recorded in our study can be correlated to severe schistosomiasis pathology. The findings of Souza et al. [76] revealed elevated MIP-1α levels in the plasma of patients with chronic schistosomiasis and decreased liver granuloma size, liver eosinophil peroxidase activity, and collagen content in *S. mansoni* infected-MIP-1α-deficient mice. Moreover, Falcao et al. [77] reported that the neutralization of MIP-1α decreased the size of *S. mansoni* pulmonary granuloma.

Interestingly, we also found that treatment with 18 mg/kg/day of PZQ for 28 consecutive days led to significant modulation of the Th1, Th2, and Th17 responses at the granulomatous stage of *S. mansoni* infection. Mice submitted to this regimen exhibited a low level of Th1, Th2 and Th17 mediated cytokines and their gene expression in the liver. In addition, high serum levels and liver gene expression of IL-10 and TGF-β, as well as an increased expression of RANTES, were also recorded. These results indicate a modulation of the immune response by this PZQ regimen. The immunoregulation by PZQ at 18mg/kg/day for 28 consecutive days is consecutive to the low granulomatous response following the reduction of adult schistosomes and tissue eggs deposition. The modulation of the immune response that is essential to promote host survival during schistosome infection can be achieved by multiple mechanisms involving IL-10. CD4$^+$CD25$^+$ Tregs and B cells are sources of IL-10. Studies with IL-10-deficient mice indicate that IL-10-producing Tregs regulate pathology by controlling both Th1 and Th2 cytokines production during *S. mansoni* infection [67,74]. Regarding the modulatory activity of RANTES, several authors reported its elevated plasma levels in patients with chronic schistosomiasis. It negatively regulates the granulomatous response and controls disease severity in *S. mansoni*-infected mice [4,67,74,76]. In addition, Huang et al. [78] demonstrated that TGF-β1 attenuates hepatic fibrosis by inhibiting the activation of PI$_3$/Akt signaling pathways. This leads to the inactivation of the proliferation and migration of hepatic stellate cells, thus inhibiting the production of ECM components in the liver of *S. japonicum*-infected mice.

Regarding the overall results of our study, among PZQ-treated groups, the treatment regimens of 100 mg/kg/day for 5 days (PZQ1) and the single dose of 500 mg/kg (PZQ4) when administered at the early stage of the infection were ineffective on *S. mansoni* pathology. On the other hand, the posologies of 100 mg/kg/day for 28 days (PZQ2) and 18 mg/kg/day for 28

days (PZQ3) were effective on the experimental mouse model of schistosomiasis. The treatment schemes mentioned above showed the best efficacy in reducing the parasitological burdens. However, critical differences were recorded between the two treatment schemes. In fact, in the PZQ3 group, the body weight, the organ indices, the hematological profile, the liver function, and the liver and spleen oxidative status were globally near-normal levels. On the contrary, despite its good schistosomicidal activity, the PZQ2 treatment scheme was ineffective on *S.mansoni*-induced liver impairment. In addition, body weight loss, anemia, and thrombocytopenia were also noticed. More importantly, the PZQ2 group presented a mortality rate of 81.25%. In this regard, the PZQ posology of 100 mg/kg/day for 28 consecutive days seems toxic for mice.

## Conclusion

Our study demonstrates that early treatment with PZQ at the dose of 18 mg/kg/day for 28 consecutive days exhibited the best antischistosomal activity. This posology considerably prevented mice against *S. mansoni* induced-pathology by reducing the parasitological burden and the hepatosplenomegaly. It also prevents hepatic and intestinal severe granulomatous reaction and fibrosis, hepatic and splenic oxidative stress, and regulates the Th1, Th2, and Th17 immune response induced by *S. mansoni* infection in mice. Therefore, we can recommend this PZQ regimen as a reference for further studies related to the antischistosomal potency of natural products or compounds in the early stage of a murine model of schistosomiasis. Upcoming studies associated with the mechanism of action will be conducted on PBMC, spleen cells and CD4 T cells harvested from *S. mansoni*-infected mice treated with PZQ at 18 mg/kg/day for 28 days to evaluate the Th1, Th2, and Th17 responses at different stages of the infection.

## Acknowledgments

We are grateful to the association "Pathologie, Cytologie et Développement" (PCD), which kindly donated equipment and reagents for histological study. We also thank Ms. Laetitia Béal and Ms. Chloé Gommenginger for their technical assistance in realizing multiplex assays.

## Author Contributions

**Conceptualization:** Ulrich Membe Femoe, Hermine Boukeng Jatsa, Catherine Cannet, Théophile Dimo, Pierre Kamtchouing, Louis-Albert Tchuem Tchuenté.

**Data curation:** Ulrich Membe Femoe.

**Funding acquisition:** Ulrich Membe Femoe, Catherine Cannet, Ahmed Abou-Bacar, Alexander Wilhelm Pfaff, Pierre Kamtchouing.

**Investigation:** Ulrich Membe Femoe, Hermine Boukeng Jatsa, Mérimé Christian Kenfack, Nestor Gipwe Feussom, Joseph Bertin Kadji Fassi, Emilenne Tienga Nkondo.

**Methodology:** Ulrich Membe Femoe, Hermine Boukeng Jatsa, Valentin Greigert, Julie Brunet, Catherine Cannet, Mérimé Christian Kenfack, Nestor Gipwe Feussom, Joseph Bertin Kadji Fassi, Emilenne Tienga Nkondo, Ahmed Abou-Bacar, Alexander Wilhelm Pfaff.

**Project administration:** Hermine Boukeng Jatsa, Alexander Wilhelm Pfaff, Louis-Albert Tchuem Tchuenté.

**Supervision:** Hermine Boukeng Jatsa, Alexander Wilhelm Pfaff, Louis-Albert Tchuem Tchuenté.

**Validation:** Hermine Boukeng Jatsa, Valentin Greigert, Catherine Cannet, Ahmed Abou-Bacar, Alexander Wilhelm Pfaff, Théophile Dimo, Pierre Kamtchouing, Louis-Albert Tchuem Tchuenté.

**Visualization:** Hermine Boukeng Jatsa, Catherine Cannet, Ahmed Abou-Bacar, Alexander Wilhelm Pfaff, Théophile Dimo, Pierre Kamtchouing.

**Writing – original draft:** Ulrich Membe Femoe.

**Writing – review & editing:** Hermine Boukeng Jatsa, Valentin Greigert, Alexander Wilhelm Pfaff.

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
