## [Decision Letter · Decision Letter 0]

3 Mar 2022

Dear Dr. Boukeng Jatsa,

Thank you very much for submitting your manuscript "Pathological and immunological evaluation of different regimens of praziquantel treatment in a mouse model of Schistosoma mansoni infection" for consideration at PLOS Neglected Tropical Diseases. As with all papers reviewed by the journal, your manuscript was reviewed by members of the editorial board and by several independent reviewers. The reviewers appreciated the attention to an important topic. Based on the two reviews, we are likely to accept this manuscript for publication, providing that you modify the manuscript according to the reviewer's comments. 

Sincerely,

John Pius Dalton, PhD

Associate Editor

Sergio Oliveira

Deputy Editor

Reviewer's Responses to Questions

**Key Review Criteria Required for Acceptance?**

**Methods**

-Are the objectives of the study clearly articulated with a clear testable hypothesis stated?

-Is the study design appropriate to address the stated objectives?

-Is the population clearly described and appropriate for the hypothesis being tested?

-Is the sample size sufficient to ensure adequate power to address the hypothesis being tested?

-Were correct statistical analysis used to support conclusions?

-Are there concerns about ethical or regulatory requirements being met?

Reviewer #1: (No Response)

Reviewer #2: I would prefer a replicated study design for any study of drug effects. However, given the complexity of the current report, I will not insist on it. I am concerned that pretesting of the dose regimens was not performed prior to treatment, as extended administration of 100 mg/kg PZQ is clearly toxic; these regimens should not have been included in experiments with infected animals. I have not checked, but surely some data on toxicology were available; the package submitted for approval would have included results from chronic testing and should be available from WHO, if not other sources. The authors should discuss this shortcoming. Is it possible that infection exacerbated the toxicity of PZQ? In addition, the authors should have included a group treated with the standard regimen of PZQ in their model. Some authors treat with a single dose of up to 500 mg/kg at 28 days, others at 42 days.

**Results**

-Does the analysis presented match the analysis plan?

-Are the results clearly and completely presented?

-Are the figures (Tables, Images) of sufficient quality for clarity?

Reviewer #1: (No Response)

Reviewer #2: No concerns about this aspect of the manuscript

**Conclusions**

-Are the conclusions supported by the data presented?

-Are the limitations of analysis clearly described?

-Do the authors discuss how these data can be helpful to advance our understanding of the topic under study?

-Is public health relevance addressed?

Reviewer #1: (No Response)

Reviewer #2: (No Response)

**Editorial and Data Presentation Modifications?**

Reviewer #1: (No Response)

Reviewer #2: (No Response)

**Summary and General Comments**

Reviewer #1: This was an interesting study of the effects of praziquantel on both the schistosome parasites and the host. The work is comprehensive and well performed. The manuscript is very easy to real and all of the figures of a high standard. There is a huge amount of data presented. While this makes the study very comprehensive, it does make the publication unwieldly as a whole. 

Line 107 please state the sex of the animals used.

I would be useful in groups could be named better ie HC, IC PZQ1-4 etc at the beginning of the line in the methods section.

The use of PZQ dissolved in distilled water is a concern, do the drugs solubility. The FDA states it is “and very slightly soluble in water”, https://www.accessdata.fda.gov/drugsatfda_docs/label/2014/018714s013lbl.pdf

Just to confirm the third group received 100mg/kg/day x 28 days, that contrasts to the other groups that in total received 500mg/kg, that 3rd group received 5 times the dosage of the others. I am not sure how that was justified, especially in the light of the very low survival rate.

In Table 2 define what the acute and granulomatous phases are.

Similarly in line 387 the terms acute stage is used but not defined until line 532. Same with granulomatous stage on line 291.

Table 1 please add references for the primers used if possible. Tables 3-5 were cut off on the right in the pdf I reviewed. I could not review these properly. I am not sure why this was not identified during the submission process. From what I can is that AST levels were elevated for the PZQ3 group? Since there were no groups where uninfected mince were given the PZQ regimes tested, there is no way to demonstrated the cytotoxic effects for the drug on the animal. But I acknowledge that the group PZQ 3 parameters are less than IC. The long term (28 days) exposure of cell lines to PZQ could have been informative.

The efficacy of a low dosage for a long period of time is clear in the many metrics provided, but what is the potential for drug resistance to develop in field settings? Drug resistance in general is not discussed.

While the results are very robust, it is not clear to me if the authors are saying if PZQ is having an effect on the host, independent to its direct antischistosomal activity. There have been some reports on the therapeutic effect of PZQ (PMID: 32606340).

I would recommend these papers be considered for your manuscript. PMID:34736899, PMID:34796025, PMID:34702326, PMID:34273315, PMID:34280206, PMID: 32741402, PMID:23555262.

Reviewer #2: The authors should discuss the possibility that PZQ bioaccumulates in chronic treatment, so that its normally short t-1/2 is extended after 28 days of dosing, especially if liver damage is present. Efficacy of 28-day regimens could be due to sufficient presence of the drug to affect worms of sufficient maturity at day 28 to be susceptible, or to previously unknown effects of prolonged duration of exposure to PZQ on immature worms. Without data on plasma levels at day 28/29, these possibilities cannot be rigorously excluded. Also, while it is true that it is desirable to prevent egg deposition in the host, daily dosing is incompatible with current control programs in the field; to prevent the development of adult parasites, people would have to receive a month of treatment every other month. It is possible that expatriates or tourists could be given a month-ling regimen of PZQ (assuming these results are relevant for humans), but it is also possible to take a normal regimen 6 weeks after leaving an endemic area to prevent pathology. The results are interesting and should be published, as they indicate that an extended duration of exposure, independent of the dose (and thus plasma levels) may be deleterious to immature schistosomes. That, I believe, is new.

PLOS authors have the option to publish the peer review history of their article (what does this mean?). If published, this will include your full peer review and any attached files.

Reviewer #1: No

Reviewer #2: No

Figure Files:

Data Requirements:

Reproducibility:

References

---

## [Editor Report · Decision Letter 1]

1 Apr 2022

Dear DR. Boukeng Jatsa,

We are pleased to inform you that your manuscript 'Pathological and immunological evaluation of different regimens of praziquantel treatment in a mouse model of Schistosoma mansoni infection' has been provisionally accepted for publication in PLOS Neglected Tropical Diseases.

Best regards,

John Pius Dalton, PhD

Associate Editor

Sergio Oliveira

Deputy Editor

---

## [Editor Report · Acceptance letter]

15 Apr 2022

Dear Pr Boukeng Jatsa,

We are delighted to inform you that your manuscript, "Pathological and immunological evaluation of different regimens of praziquantel treatment in a mouse model of Schistosoma mansoni infection," has been formally accepted for publication in PLOS Neglected Tropical Diseases.

Best regards,

Shaden Kamhawi

co-Editor-in-Chief

Paul Brindley

co-Editor-in-Chief
